# Nuclear lamina integrity is required for proper spatial organization of chromatin in *Drosophila*

Sergey V. Ulianov[1,2], Semen A. Doronin[3], Ekaterina E. Khrameeva [4,5], Pavel I. Kos[6], Artem V. Luzhin[1],
Sergei S. Starikov[7], Aleksandra A. Galitsyna[1,4,5,7], Valentina V. Nenasheva[3], Artem A. Ilyin[3], Ilya M. Flyamer[8],
Elena A. Mikhaleva[3], Mariya D. Logacheva[4,9,10], Mikhail S. Gelfand[4,5,11], Alexander V. Chertovich[6],
Alexey A. Gavrilov[1], Sergey V. Razin[1,2] & Yuri Y. Shevelyov[3]

How the nuclear lamina (NL) impacts on global chromatin architecture is poorly understood. Here, we show that NL disruption in *Drosophila* S2 cells leads to chromatin compaction and repositioning from the nuclear envelope. This increases the chromatin density in a fraction of topologically-associating domains (TADs) enriched in active chromatin and enhances interactions between active and inactive chromatin. Importantly, upon NL disruption the NL-associated TADs become more acetylated at histone H3 and less compact, while background transcription is derepressed. Two-colour FISH confirms that a TAD becomes less compact following its release from the NL. Finally, polymer simulations show that chromatin binding to the NL can per se compact attached TADs. Collectively, our findings demonstrate a dual function of the NL in shaping the 3D genome. Attachment of TADs to the NL makes them more condensed but decreases the overall chromatin density in the nucleus by stretching interphase chromosomes.

[1] Institute of Gene Biology, Russian Academy of Sciences, Moscow 119334, Russia. [2] Faculty of Biology, M.V. Lomonosov Moscow State University, Moscow 119991, Russia. [3] Institute of Molecular Genetics, Russian Academy of Sciences, Moscow 123182, Russia. [4] Skolkovo Institute of Science and Technology, Skolkovo 143026, Russia. [5] Institute for Information Transmission Problems (the Kharkevich Institute), Russian Academy of Sciences, Moscow 127051, Russia. [6] Faculty of Physics, M.V. Lomonosov Moscow State University, Moscow 119991, Russia. [7] Faculty of Bioengineering and Bioinformatics, M.V. Lomonosov Moscow State University, Moscow 119991, Russia. [8] MRC Human Genetics Unit, Institute of Genetics and Molecular Medicine, University of Edinburgh, Edinburgh EH4 2XU, UK. [9] Belozersky Institute of Physico-Chemical Biology, M.V. Lomonosov Moscow State University, Moscow 119234, Russia. [10] Russia Extreme Biology Laboratory, Institute of Fundamental Medicine and Biology, Kazan Federal University, Kazan 420012, Russia. [11] Faculty of Computer Science, National Research University Higher School of Economics, Moscow 125319, Russia. These authors contributed equally: Sergey V. Ulianov, Semen A. Doronin, Ekaterina E. Khrameeva, Pavel I. Kos. Correspondence and requests for materials should be addressed to S.V.U. (email: sergey.v.ulyanov@gmail.com) or to Y.Y.S. (email: shevelev@img.ras.ru)

The nuclear lamina (NL)[1] is a meshwork of lamins and lamin-associated proteins lining the nuclear envelope (NE). Several lines of evidence support the idea that the NL is a platform for the assembly of the repressive compartment in the nucleus. In mammals, nematode and *Drosophila*, the lamina-associated chromatin domains (LADs)[2–5] contain mostly silent or weakly expressed genes[2–6]. Activation of tissue-specific gene transcription during cell differentiation is frequently associated with translocation of loci from the NL to the nuclear interior[4,7–11]. The expression level of a reporter gene is ~5-fold lower when it is inserted into LADs compared to inter-LADs[12]. Artificial tethering of weakly expressed reporter genes to the NL results in their downregulation thus indicating that contact with the NL may cause their repression[13–15]. Accordingly, many transcriptional repressors, including histone deacetylases (HDACs) are linked to the NL[16]. The high throughput chromosome conformation capture (Hi-C) technique has revealed the spatial segregation of open (DNase I-sensitive) and closed (DNase I-resistant) chromatin into two well-defined compartments[17]. Importantly, in mammalian cells, the DNase I-resistant compartment is strongly enriched with NL contacts[18,19]. Moreover, a whole-genome DNase I-sensitivity assay in *Drosophila* S2 cells indicated that LADs constitute the densely packed chromatin[20]. Additionally, super-resolution microscopy studies in Kc167 cells show that inactive chromatin domains (including Polycomb (Pc)-enriched regions) are more compact than active ones[21].

The newly developed single-cell techniques demonstrate that LADs, operationally determined in a cell population, may be located either at the NL or in the nuclear interior in individual cells[19,22]. Surprisingly, the positioning of LADs in the nuclear interior barely affects the inactive state of their chromatin[22]. This raises the question as to whether contact with the NL makes the chromatin in LADs compact and inactive. However, few studies directly address this issue. It has been shown that lamin Dm0 knock-down (Lam-KD) in *Drosophila* S2 cells decreases the compactness of a particular inactive chromatin domain[23]. Accordingly, the accessibility of heterochromatic and promoter regions has been shown to increase upon Lam-KD in *Drosophila* S2R[+] cells[24]. However, the impact of the NL on the maintenance of the overall chromatin architecture remains mostly unexplored.

Here we show that upon loss of all lamins, the density of peripheral chromatin is decreased in *Drosophila* S2 cells leading to the slight overall chromatin compaction. At the same time, chromatin in LADs becomes less tightly packed which correlates with the enhancement of initially weak level of histone H3 acetylation and background transcription in these regions.

## Results

### Lam-KD in S2 cells results in general chromatin compaction.

We have studied the effects of NL disruption on global chromatin architecture, histone acetylation and gene expression in *Drosophila*. To select an appropriate experimental model, we first analysed the presence of ubiquitous lamin Dm0 and tissue-specific lamin C proteins[25] in several *Drosophila* cell lines by Western-blotting. Whereas the level of lamin Dm0 is similar in S2, Kc167, and OSC lines, lamin C is robustly present in Kc167 and OSC, but almost completely absent in S2 cells (Fig. 1a). Hence, to remove all lamins, we performed Lam-KD in S2 cells by RNAi (Fig. 1b) and stained the nuclei with anti-histone H4 antibody to visualise the bulk chromatin, and with anti-lamin-B-receptor (LBR[26]) antibody to visualise the NE (Fig. 1c and Supplementary Fig. 1a). Quantification of the fluorescence intensity along the nuclear diameter reveals a slight but statistically significant shift in the radial distribution of total chromatin from the NE towards the nuclear interior upon Lam-KD (Fig. 1d and

Supplementary Fig. 1a). To validate this observation, we performed fluorescence in situ hybridization (FISH) with a probe from the cytological region *36C*, which was previously mapped as a LAD in the Kc167 cells[5] (Fig. 1e). The radial position of this region is shifted towards the nuclear interior in Lam-KD S2 cells when compared to control cells (hereinafter treated with dsRNA against bacterial *lacZ* gene) (Fig. 1e). Notably, this observation agrees with previously published results[11] which we reanalysed to demonstrate a shift in the radial position of two other loci (*22A* and *60D*) from the NE upon Lam-KD (Fig. 1f). Moreover, we observed an *en masse* chromatin compaction as a result of NL disruption, since the average volume of total chromatin, reconstructed by DAPI staining, is markedly diminished upon Lam-KD (Fig. 1g and Supplementary Fig. 1b). Remarkably, the average volume of nuclei, reconstructed by LBR-stained NE, was not affected by Lam-KD (Supplementary Fig. 1c). Taken together, these observations indicate that disruption of the NL results in general chromatin compaction and repositioning from the NE.

In mammalian cells, the presence of either lamin A/C or LBR is necessary for proper positioning of the heterochromatin at the nuclear periphery[27]. In contrast, in *Drosophila* S2 cells, where lamin C is not expressed (Fig. 1a), depletion of LBR does not notably affect chromatin positioning relative to the NE (Fig. 1h and Supplementary Fig. 1d). We confirmed this observation by FISH with the probe from the *36C* region examined upon Lam-KD. We found that this region is not repositioned relative to the NE upon LBR depletion (Fig. 1i). These results indicate that the main heterochromatin tethers are different in mammals and *Drosophila* with the lamin Dm0 providing the major impact on LAD attachment, at least in S2 cells.

### Lam-KD in S2 cells enhances weak transcription in LADs.

To examine which genes are associated with the NL, we have used previously published lamin–DamID data for Kc167 cells[5,28] that are closely related to S2 cells, are of a similar embryonic origin, and have highly correlated transcriptome profiles (Pearson's $R = 0.89$)[29]. As we were not confident that LADs on the X chromosome occupy the same positions in the female Kc167 and in the male S2 cells[30], we excluded X-chromosome from the downstream analysis. We hypothesised that upon Lam-KD, the detachment of LADs from the NE might result in the elevated expression of genes located therein. To test this hypothesis, we performed transcriptome profiling in control and Lam-KD S2 cells using RNA-seq (Supplementary Fig. 2a) and revealed 60 differentially expressed genes (40 up- and 20 downregulated genes) (Fig. 2a). However, the observed increase in gene expression (Supplementary Fig. 2b) does not correlate with the presence of promoters of differentially expressed genes specifically in LADs ($P = 0.21$, permutation test), thus suggesting that either an indirect effect of NL disruption or alterations in chromatin interactions in the nuclear interior are affecting transcription. We then analysed changes in total transcription inside and outside of LADs (i.e. in the inter-LADs). Depletion of lamin Dm0 results in the moderate upregulation of the generally very weak background transcription in LADs (Supplementary Fig. 2c), but not in the inter-LADs (Fig. 2b, c).

To confirm RNA-seq results, we applied RT-qPCR to analyse the transcription level of 14 randomly selected genes whose promoters are located in different LADs (Supplementary Table 1). Almost all of these genes are expressed in S2 cells at a very low level. 12 out of 14 genes appeared to be upregulated upon Lam-KD (~2 fold on average) when compared to control S2 cells (Fig. 2d, top panel). It has previously been shown that Lam-KD in *Drosophila* S2 cells results in increased DNase I sensitivity and the derepression of several testis-specific genes in the silent chromatin domain from the *60D* chromosomal region[11]. We

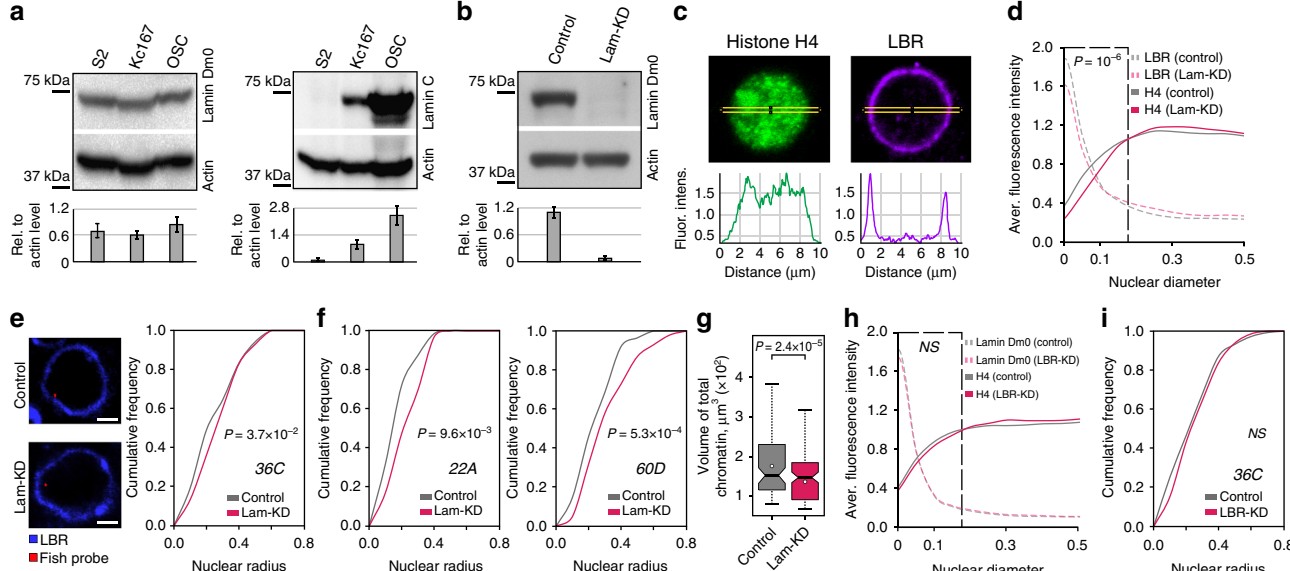

**Fig. 1** Chromatin is released from the NE and becomes denser upon Lam-KD in S2 cells. **a**, **b** Western-blot analysis of lamin Dm0 and lamin C protein levels in *Drosophila* cell lines (**a**), or in Lam-KD and control S2 cells (**b**). Band intensity quantitation is presented below. **c** A representative example of nuclei immunostained with antibodies against histone H4 and LBR. Fluorescence intensity along the yellow-framed zone was measured using ImageJ software. **d** Averaged fluorescence intensity profiles along the nuclear diameter in Lam-KD ($n = 120$) and control ($n = 180$) S2 cells immunostained with antibodies against histone H4 and LBR. $P$ value for the framed region of histone H4 profiles was estimated in a Wilcoxon test. **e** Cumulative frequency of radial positions of the FISH probe located in the LAD from cytological region *36C* in Lam-KD ($n = 175$) or control ($n = 175$) S2 cells (right panel). $P$ value was estimated in a Kolmogorov-Smirnov one-sided test. Representative examples of FISH signals in nuclei stained with anti-LBR antibody are shown on the left panels. Scale bar 1 μm. **f** Cumulative frequency of radial positions of FISH probes to cytological regions *22A* (left) or *60D* (right) in Lam-KD ($n = 100$ for *22A* and $n = 99$ for *60D*) or control ($n = 115$ for *22A* and $n = 72$ for *60D*) S2 cells. $P$ value was estimated in a Kolmogorov-Smirnov one-sided test. Data for analysis were taken from ref. [11]. **g** Total chromatin volume measured by DAPI fluorescence in Lam-KD ($n = 125$) or control ($n = 83$) S2 cells. $P$ value was estimated in a Wilcoxon test. Thick black lines and white dots represent median and average values, upper and lower ends of boxplot show the upper and lower quartiles, the whiskers indicate the maximum and minimum values. **h** Averaged fluorescence intensity profiles along the nuclear diameter in LBR-depleted (LBR-KD, $n = 120$) or control ($n = 120$) S2 cells immunostained with antibodies against histone H4 and lamin Dm0. *NS* – non-significant difference ($P > 0.05$, Wilcoxon test). **i** Cumulative frequency of radial positions of the FISH probe to cytological region *36C* in LBR-KD ($n = 150$) or control ($n = 150$) S2 cells. *NS* – non-significant difference ($P > 0.05$, Kolmogorov-Smirnov one-sided test)

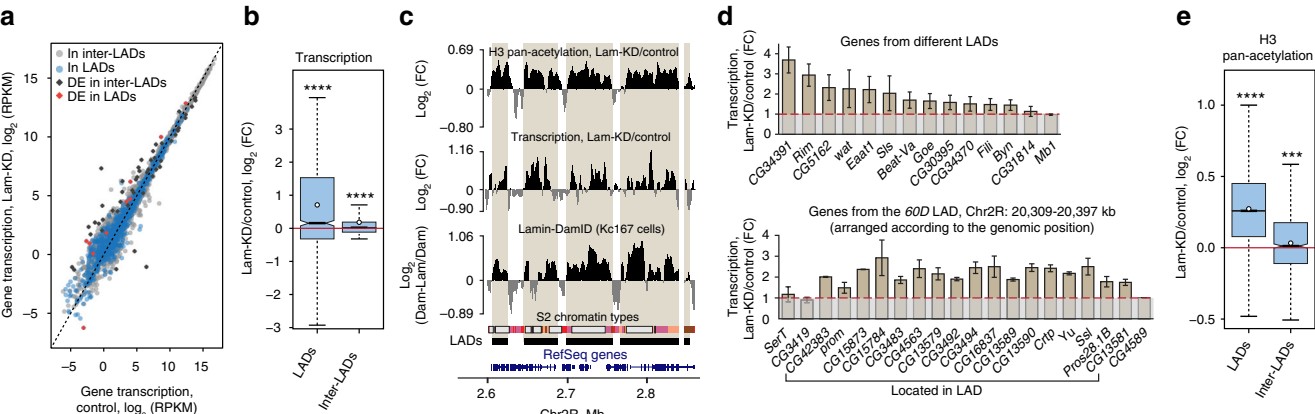

**Fig. 2** Nuclear lamina disruption leads to the increased H3 pan-acetylation and transcriptional upregulation in LADs. **a** Gene expression in Lam-KD and control S2 cells according to the RNA-seq data. Genes located in LADs and inter-LADs are designated by light-blue and light-grey circles, respectively. Differentially expressed (DE) genes, located in LADs and inter-LADs, are designated by black and red rectangles, respectively. **b** Changes of total transcription according to the RNA-seq data between Lam-KD and control S2 cells in LADs and in inter-LADs. **c** A representative screenshot from UCSC Genome Browser showing the Lamin–DamID profile in Kc167 cells[5] and log₂(FC) profiles of H3 pan-acetylation (according to ChIP-seq) and transcription (according to RNA-seq) in Lam-KD/control S2 cells. Chromatin annotation in S2 cells[40], LAD annotation in Kc167 cells[28], and RefSeq gene positions are shown below. **d** Changes in the transcription level in Lam-KD compared to control S2 cells by RT-PCR analysis for the randomly chosen genes from different LADs (top panel), and for all the genes from the *60D* LAD (bottom panel). In the bottom panel, genes *SerT, CG3419* and *CG4589* are located outside the LAD. Expression data of *Crtp, Yu, Ssl, Pros28.1B* and *CG13581* genes upon Lam-KD in S2 cells are from ref. [11]. Error bars show SEM between two independent biological replicates. **e** Changes of H3 pan-acetylation according to ChIP-seq data between Lam-KD and control S2 cells in LADs and in inter-LADs. In panels (**b**, **e**), ****$P < 0.0001$, ***$P < 0.001$ in a Wilcoxon test. See Fig. 1g legend for description of boxplot elements

found that all the genes located in this LAD are almost uniformly upregulated upon Lam-KD in S2 cells (Fig. 2d, bottom panel). Thus, NL disruption results in the partial derepression of chromatin in LADs leading to the increased background transcription.

It has been shown that pan-acetylation of histones H3 and H4 coupled with general DNase I-sensitivity was elevated in the *60D* LAD upon Lam-KD in S2 cells[23]. To check whether the repression of transcription in LADs may be caused by histone deacetylation, we determined histone H3 pan-acetylation level across the entire genome by chromatin immunoprecipitation (ChIP-seq) (Supplementary Fig. 2d). We found that the general level of histone H3 acetylation is markedly elevated in LADs, but not in the inter-LADs upon Lam-KD when compared to control cells (Fig. 2c, e). Thus, we suggest that a fraction of HDACs associated with the NL[31,32] may be at least partially responsible for the low level of histone H3 acetylation and for the transcriptional repression in LADs making their chromatin less accessible for spurious binding by *trans*-acting factors.

**Lam-KD in S2 cells leads to decompaction of inactive TADs.** To explore genome-wide effects of NL disruption on the spatial organization of chromatin, we applied the Hi-C technique[17]

to control and Lam-KD S2 cells and identified topologically-associating domains (TADs)[33–35] (Fig. 3a) using previously described two-step procedure[36]. The strong similarity between Hi-C map data obtained in this work with that previously published for S2 cells[37] (Supplementary Fig. 3a), as well as the high correlation between Hi-C replicates (Supplementary Fig. 3b) demonstrates the high quality and reliability of the data. Furthermore, in agreement with the conservation of TAD boundaries in unrelated *Drosophila* cell types[36,38] and upon different biological conditions[39], pairwise comparison of TAD positions between Lam-KD and control cells does not show statistically significant alterations (Supplementary Fig. 3c). We conclude that NL disruption in S2 cells does not affect the overall TAD profile genome-wide. This allows us to compare the average contact frequency (ACF, see Methods) within each TAD between control and Lam-KD S2 cells. We argue that differences in ACF should reflect changes in the physical density of a TAD.

Figure 3b shows a clear negative trend between LAD coverage within a TAD and intra-TAD ACF changes upon Lam-KD relative to control cells. This trend is absent when LAD coverage is plotted against ACF variability between control replicates (Supplementary Fig. 3d). Remarkably, the opposite trend is revealed between intra-TAD ACF changes and the proportion

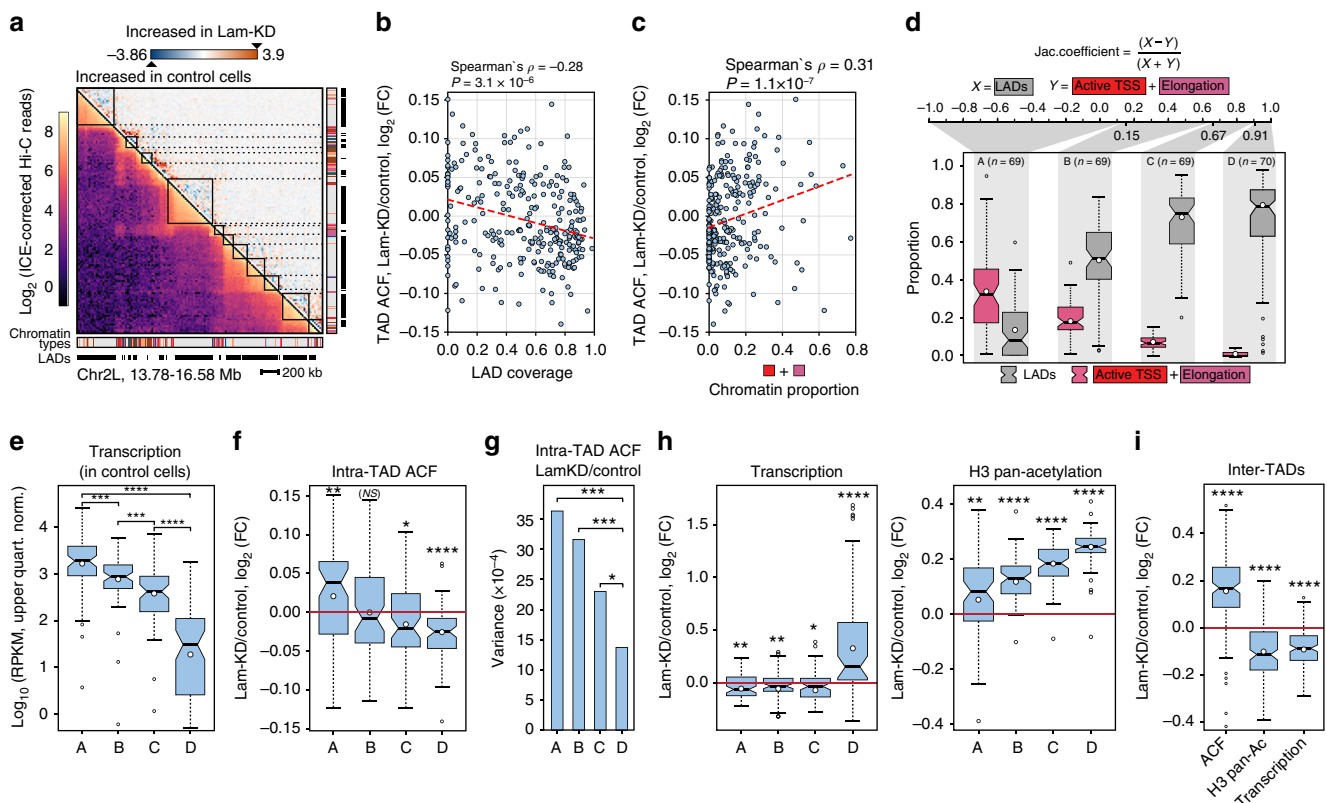

**Fig. 3** TADs respond differentially to NL disruption. **a** Hi-C map showing a total chromatin interaction profile (left half of the map), or subtraction map (Lam-KD - control) (right half of the map) at a 2.8-Mb region of chromosome 2 L. Chromatin annotation in S2 cells[40] and LADs annotation in Kc167 cells[28] are shown below. **b, c** Dependence of intra-TAD ACF changes upon Lam-KD on the LAD coverage (**b**), or on the proportion of "red" and "purple" chromatin types[40] (**c**) within these TADs. Trend line is in red. **d** Separation of TADs into four groups according to the Jaccard similarity coefficient. Box plots show the proportion of active chromatin ("red" (active TSS) plus "purple" (elongation) chromatin types[40]) and LAD coverage[28] within each group. **e** Transcription level in the four groups of TADs according to RNA-seq in control S2 cells. **f** Changes of intra-TAD ACF between Lam-KD and control cells in the four groups of TADs. **g** Variances of log₂(FC) of the intra-TAD ACF upon Lam-KD compared to control cells in the four groups of TADs. ***$P < 0.001$, *$P < 0.05$ in a Levene's test. **h** Changes of total transcription (left panel) and H3 pan-acetylation (right panel) levels between Lam-KD and control cells in the four groups of TADs. **i** Changes of ACF values, H3 pan-acetylation and total transcription in the inter-TAD regions between Lam-KD and control cells. See Fig. 1g legend for description of boxplot elements represented on panels (**d–f**), (**h**) and (**i**). In panels **e, f, h, i**, ****$P < 0.0001$, ***$P < 0.001$, **$P < 0.01$, *$P < 0.05$, NS non-significant difference ($P > 0.05$) in a Wilcoxon test

of "red" plus "purple", but not "coral" plus "brown" active chromatin types (according to 9-type chromatin annotation in S2 cells[40]; Fig. 3c and Supplementary Fig. 3e). To simultaneously account for the LAD coverage and the proportion of "red" plus "purple" chromatin types, we calculated the Jaccard coefficient between these two metrics for each TAD. Based on this, we then divided the ranked TADs into four equal-sized groups A, B, C and D (Fig. 3d), with TADs in group A being relatively enriched in active chromatin and depleted of LADs, and TADs in group D being depleted of active chromatin and enriched in LADs. Consistent with chromatin type annotation, transcription level in control S2 cells appears to be highest in TADs from group A, and lowest in TADs from group D (Fig. 3e).

Strikingly, we observed the opposite changes of ACF values upon Lam-KD in the TADs from groups A and D having polar metrics. ACF values are increased in the TADs from group A containing the highest proportion of active chromatin and the lowest LAD coverage and are decreased in the TADs from group D with the lowest proportion of active chromatin and the highest LAD coverage (Fig. 3f). TADs from groups B and C, which preserve (group B) or slightly decrease (group C) their ACFs upon Lam-KD (Fig. 3f), likely represent the mixture of chromatin increasing and decreasing its density. In support of this idea, the variance of ACF changes is the lowest within group D TADs (Fig. 3g) which strongly correspond to LADs, when compared to other groups containing the mixture of active and inactive chromatin types (Fig. 3d).

Consistent with the transcriptional derepression in LADs (Fig. 2a, b), the overall level of transcription is markedly elevated in the group D TADs upon Lam-KD when compared to control cells (Fig. 3h, left panel). In contrast, TADs from group A demonstrate a weak decrease in transcription upon Lam-KD. Moreover, we found that upon Lam-KD, the histone H3 acetylation level is enhanced in TADs in a strong quantitative manner dependent on their LAD coverage (Supplementary Fig. 3f), with the most pronounced increase of acetylation observed in the group D TADs (Fig. 3h, right panel).

We then asked how Lam-KD influences ACF, transcription and histone H3 acetylation in the inter-TADs which represent the most active genome regions[36]. In agreement with the observations for the group A TADs, we found an increase of ACF and a decrease in transcription within inter-TAD regions upon Lam-KD (Fig. 3i). However, contrary to group A TADs, total histone H3 acetylation level appears to be decreased upon Lam-KD in the inter-TADs (Fig. 3i).

Collectively, these findings indicate that upon NL disruption, chromatin becomes more densely packed in the active, and less densely packed in the inactive genomic regions.

**Lam-KD in S2 cells impairs spatial chromatin segregation.** In mammals, TADs belonging to the same epigenetic type (active or inactive) tend to interact with each other across large genomic distances, thus partitioning the interphase chromatin into A and B compartments[17]. The molecular mechanisms driving such interactions are largely unknown, but a role for the NL has been suggested[6]. To identify chromatin compartments in control and Lam-KD S2 cells, we applied principal component analysis (PCA), which is commonly used for compartment calling[17]. The first principal component (PC1) profile clearly correlates with the transcription profile, where the positive values of PC1 correspond to the transcriptionally active loci (Fig. 4a and Supplementary Fig. 4a and b). Surprisingly, and contrary to the findings in *Drosophila* embryos[35], the spatially distant interactions in control S2 cells appear to be enhanced, relative to those expected, for only the genomic regions with a PC1 > 0, i.e. within the active A compartment (Fig. 4b). To verify that this is not due to technical problems in our analysis, we applied PCA to the previously published Hi-C data[35] and confirmed the existence of A and B compartments in the embryos (Supplementary Fig. 4c). Upon Lam-KD in S2 cells, interaction frequency is markedly decreased within the A compartment and is increased for the regions with a PC1 < 0, i.e. within the inactive chromatin (Fig. 4c and d and Supplementary Fig. 4d). Importantly, we observed the notable gain of interactions between A compartment and the rest of the genome upon Lam-KD (Fig. 4d). These results indicate that NL disruption leads to partial "blurring" of chromatin compartmentalisation.

**Chromatin density is decreased upon TAD release from the NE.** One of the most striking observations of our study is the decrease of chromatin density in a fraction of NL-attached TADs upon their release from the NE. To confirm this observation by an alternative approach, we performed two-colour FISH using a pair of probes positioned at the borders of a long TAD/LAD which is located at the cytological region *36C* and which reduces its ACF upon Lam-KD (Fig. 5a). We found that the inter-probe distances in this TAD normalised to the nuclear radius (radial-normalised distance, RND) are increased upon Lam-KD compared to those in control cells (Fig. 5b), thus indicating that the chromatin density of this TAD decreases. Moreover, in a fraction

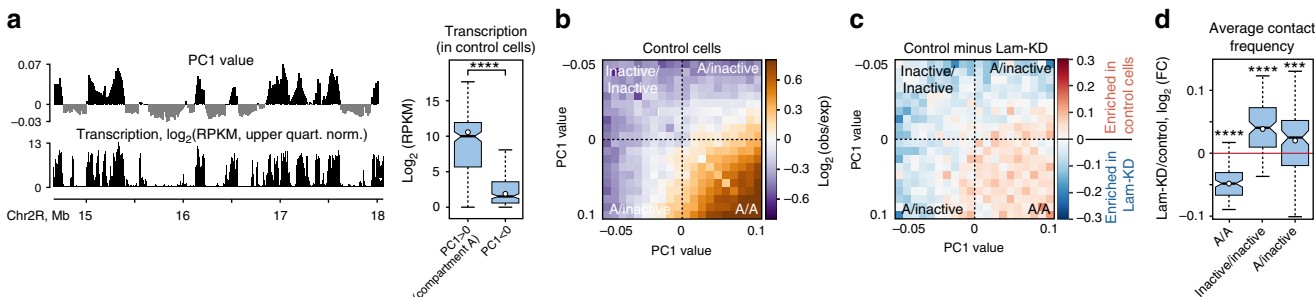

**Fig. 4** NL disruption partially impairs spatial segregation of active and inactive chromatin. **a** A representative screenshot from the UCSC Genome Browser showing correspondence between the positive PC1 values and the transcriptionally active genome regions. Box plots show transcription level in 20-kb genomic bins possessing positive (n = 4073) and negative (n = 1868) PC1 values. **** – P < 0.0001 in a Wilcoxon test. **b** Heatmap showing log₂ values of contact enrichment for the intra-chromosomal contacts in the control cells between genomic regions as a function of their PC1 values (saddle plot). **c** Subtraction of saddle plots in Lam-KD cells from that in control cells. **d** Changes of averaged contact frequency between and within active and inactive fractions of chromatin in Lam-KD compared to control cells. ****P < 0.0001, ***P < 0.001 in a Wilcoxon test. See Fig. 1g legend for description of boxplot elements represented on panels (**b**) and (**d**)

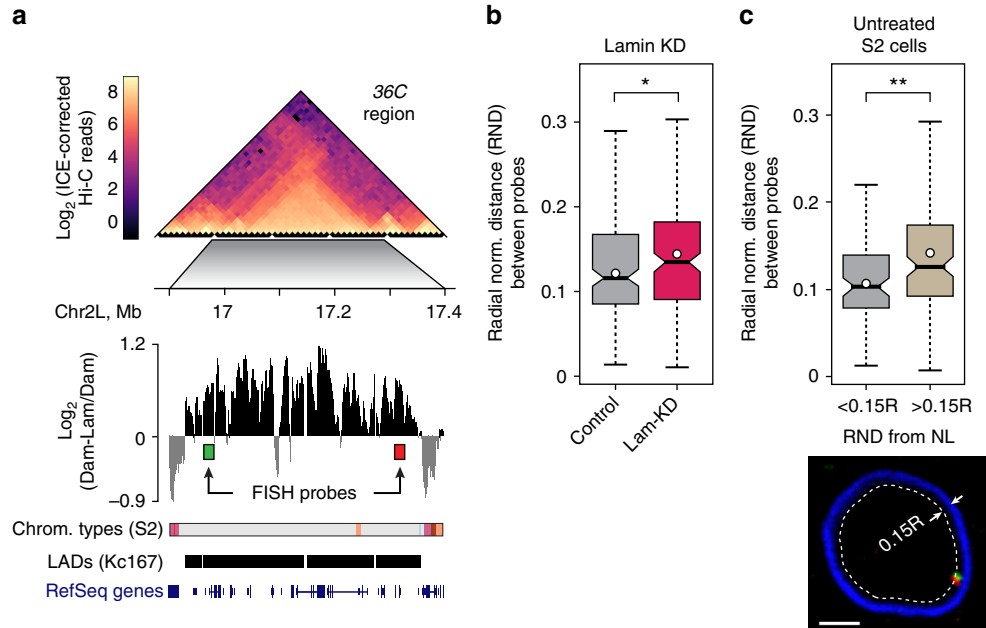

**Fig. 5** Chromatin density is decreased in a TAD following its detachment from the NE. **a** Genomic region *36C* carrying probes for two-colour FISH analysis. The Hi-C map from untreated S2 cells, lamin–DamID profile from Kc167 cells[5], the occupancy of 9 chromatin types from S2 cells[40], LAD annotation from Kc167 cells[28], and RefSeq gene positions are shown. **b** Radial-normalised distance (RND) between two FISH probes in Lam-KD (n = 175) and control (n = 175) S2 cells. *P < 0.05 in a Kolmogorov-Smirnov one-sided test. **c** RND between two FISH probes in the untreated S2 cells. RND < 0.15, n = 84, RND > 0.15, n = 124. **P < 0.01 in a Kolmogorov-Smirnov one-sided test. A representative example of two-colour FISH signals (red and green) in a nucleus stained with anti-LBR antibody is shown below. Nuclear shell of 0.15 R width (R nuclear radius) where the signals are in visible contact with the NL is outlined by the dotted line. Scale bar 1 μm. See Fig. 1g legend for description of boxplot elements represented on panels (**b**, **c**)

of untreated S2 cells, where both FISH probes are confined within the shell adjoining to the NL, the RND between probes are smaller than between probes located more distally from the NE (Fig. 5c). Thus, detachment from the NL appears to be sufficient for LAD decompaction even if the NL is intact.

**NL is able to mechanically compact LADs**. To find out whether it is an inherent feature of the chromatin in LADs to become loosely packed after the detachment from the NL, we performed polymer modelling of chromatin-NE interactions. We employed dissipative particle dynamics (DPD)-based simulation of a model polymer (MP) whose folding pattern closely recapitulates formation of globular chromatin domains (such as TADs and LADs) built up from non-acetylated nucleosomes[36]. Here, the MP folding is simulated in the presence of a surface mimicking the NL due to its ability to interact with globular domains of the MP (blue blocks, containing "non-acetylated" sticky particles, Fig. 6a). Each blue block of the MP adopts two alternative states: it can be considered as a LAD when attached to the surface by at least one particle, or as a non-LAD when none of its particles are in contact with the surface (Fig. 6a). To obtain a dataset large enough for statistical analysis, we performed ten independent simulations. Firstly, we observe a clear TAD profile at the ensemble distance map indicating that the presence of a surface does not influence the overall folding pattern of the MP (Supplementary Fig. 5). We then ranked blue blocks from all runs according to the number of their contacts with the surface and plotted these values against the number of spatial interactions between particles in each block. We revealed a positive correlation between the number of intra-block contacts and the number of particles within this block interacting with the surface (Fig. 6b). Accordingly, the volume of a block decreases (Fig. 6c) and the shape of a block gradually changes from a sphere to a "pancake" with an increasing number

of surface contacts (Fig. 6d). These results indicate that chromatin attachment to the NL per se is sufficient to compact LADs, likely confining interactions between nucleosomes in a LAD from a 3D volume to a 2D surface.

## Discussion
Here, using a variety of approaches we explored what happens to chromatin upon NL disruption. Using immunostaining and FISH experiments, we revealed that Lam-KD in *Drosophila* S2 cells leads to a slight reduction in total chromatin volume and, as a result, an increase in chromatin packaging density (Fig. 1 and Supplementary Fig. 1). However, the stronger compaction of chromatin is not homogeneous and depends on the epigenetic state and scale. Our Hi-C analysis clearly indicates two opposite trends in chromatin behaviour. The contact frequency in the active chromatin increases over short distances (i.e. within the "active" TADs and the inter-TADs) and decreases over long distances (i.e. within the A compartment). Whereas in the inactive chromatin it, inversely, decreases over short distances (i.e. within the TADs mostly corresponding to LADs), but increases at the chromosomal scale (Figs. 3 and 4).

We suggest a model explaining general chromatin stretching as well as the condensation of inactive chromatin in TADs mediated by the NL (Fig. 6e). If chromatin mobility is constrained by its tethering to the NL, then the release from this tethering will lead to chromatin shrinkage due to macromolecular crowding[41] and inter-nucleosomal interactions[42,43]. Therefore counterintuitively, the NL appears not to restrict chromatin expansion but provides an anchoring surface necessary to keep interphase chromosomes slightly stretched. At the same time, inactive chromatin may become additionally condensed due to the deacetylation by HDACs, linked to the NL[31,32], and/or mechanically, due to chromatin binding with the NL (Fig. 6d).

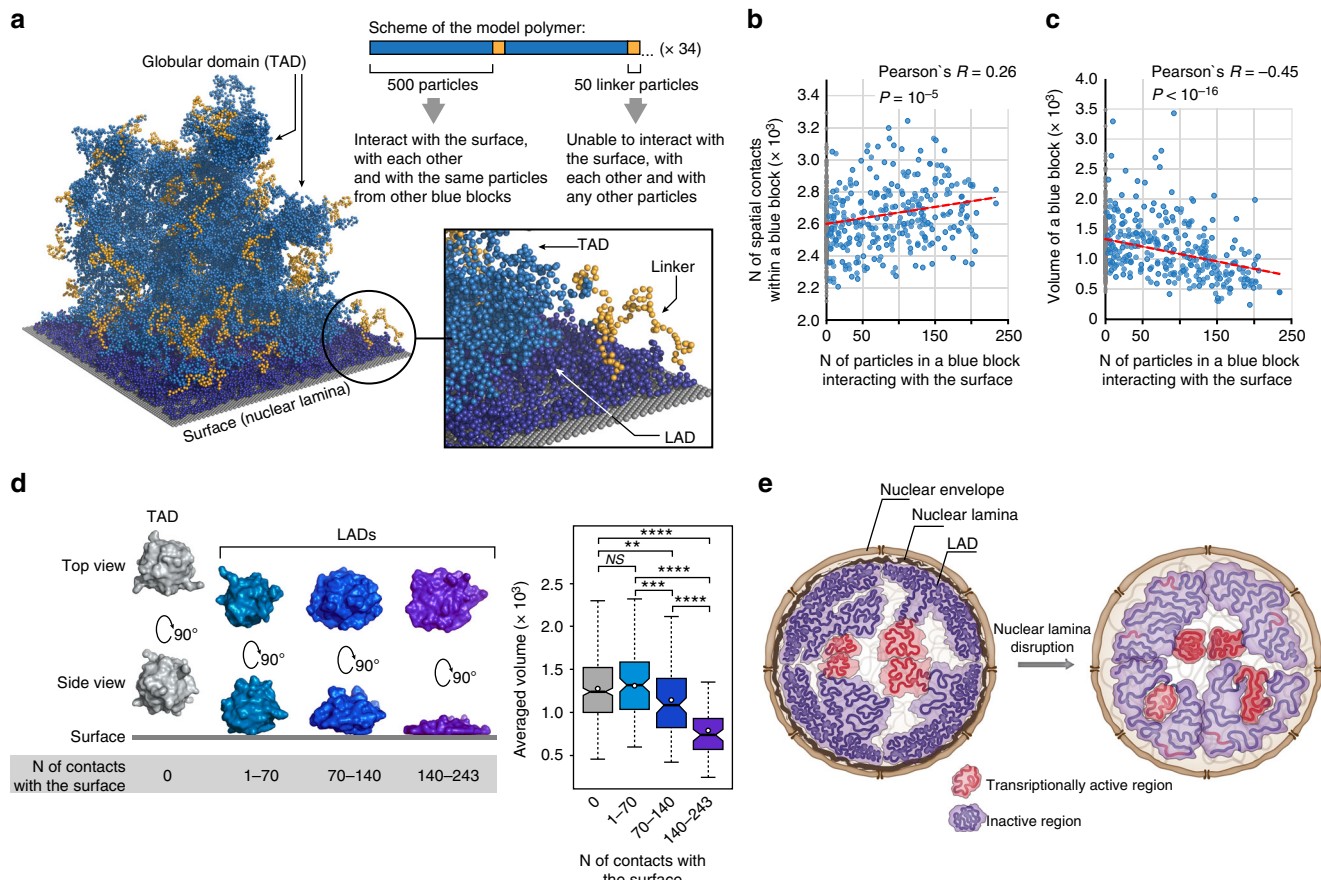

**Fig. 6** Model polymer simulation shows that the attachment to the NL is sufficient for chromatin compaction. **a** Scheme of the model polymer (MP) and visualisation of its conformation derived from a simulation run. Blue particles are able to interact with each other and with the surface recapitulating properties of non-acetylated nucleosomes in the inactive chromatin. Orange particles are unable to interact with any particles and with the surface recapitulating properties of acetylated nucleosomes in active chromatin. MP is composed of 68 blue and 68 orange particle blocks. Blue blocks with at least one particle contacting the surface are coloured with dark blue and considered as LADs. **b** The number of spatial contacts between particles in a blue block depends positively on the number of particles in this block contacting the surface. Trend line is in red. **c** The volume of a blue block depends inversely on the number of particles in this block contacting the surface. Trend line is in red. **d** Averaged 3D structures of blue blocks with a different number of contacts with the surface (left panel). Box plots (see Fig. 1g legend for description of their elements) show the decrease of LAD volume with the increasing number of contacts with the surface (right panel). ****$P < 0.0001$, ***$P < 0.001$, **$P < 0.01$, NS non-significant difference ($P > 0.05$) in a Wilcoxon test. **e** Illustrative representation of key observations made in this work

A recently published study analysed the 3D genome organization upon NL disruption in mouse embryonic stem cells (mESCs)[44]. It is interesting to compare our results from *Drosophila* with those from mice. Upon loss of all lamins, the general TAD profile is still preserved in both species, however, intra- and inter-TAD interactions are altered. Strikingly, upon loss of all lamins, a fraction of NL-attached TADs becomes less condensed in both species (Fig. 3; ref. [44]). However, in contrast to *Drosophila* S2 cells, this is not accompanied by a general detachment of chromatin from the NE in mESCs[44]. Additionally, distant interactions within the inactive chromatin are mostly increased in both species upon lamin loss (Fig. 4d; ref. [44].). Finally, while some genes located at the nuclear periphery and in the nuclear interior have changed their expression both in mESCs[44] and in *Drosophila* S2 cells (Fig. 2a), an increase in the background transcription upon lamin loss is detected specifically in *Drosophila* LADs (Fig. 2b), and this was not reported for mESCs[44]. Taken together, these findings indicate that both in mammals and *Drosophila* the NL not only makes nearby chromatin more compact and repressed, but also affects chromatin interactions and gene expression in the nuclear interior.

The diversity of mechanisms of chromatin attachment to the NL in *Drosophila* and mammals may explain the differences in chromatin behaviour in response to the lack of all lamins. For example, it was shown that LBR and PRR14 proteins participate in the tethering of the H3K9-methylated chromatin to the NE in mammals[27,45]. Whereas in mammalian ESCs LADs are strongly enriched with the H3K9me2/3[2,4,46], in *Drosophila* Kc167 and, likely, in S2 cells this modification is not present in LADs[5,28]. Accordingly, our results indicate that LBR is not required to keep chromatin at the nuclear periphery in S2 cells (Fig. 1h, i). Therefore, the removal of all lamins may not be sufficient to detach all LADs from the NE in mESCs, but can release LADs in *Drosophila* S2 cells.

In conclusion, using different approaches we revealed that NL disruption in *Drosophila* S2 cells leads to general chromatin compaction, accompanied by the impaired spatial segregation of total chromatin into active and inactive types, and the decompaction of a fraction of NL-attached TADs linked to partial derepression of their chromatin. Importantly, the observed phenomena may be related to the abnormal expression of genes in lamin-associated diseases[1].

## Methods

**Cell cultures and RNAi.** The *Drosophila melanogaster* S2 cell line (from the collection of IMG RAS) and Kc167 cell line (from the Drosophila Genomics Resource Center) were grown at 25 °C in Schneider's Drosophila Medium (Gibco) supplemented with 10% heat-inactivated fetal bovine serum (FBS, Gibco), 50 units/ml penicillin, and 50 μg/ml streptomycin. OSCs[47] kindly provided by M. Siomi were cultured in Shields and Sang M3 insect medium (Sigma-Aldrich) supplemented with 10% heat-inactivated FBS (Gibco), 10% fly extract (http://biology.st-andrews.ac.uk/sites/flycell/flyextract.html), 10 μg/ml insulin (Sigma-Aldrich), 0.6 mg/ml glutathione (Sigma-Aldrich), 50 units/ml penicillin, and 50 μg/ml streptomycin. dsRNAs against lamin *Dm0* for RNAi treatment of S2 cells were prepared as previously described[11]. dsRNAs against *LBR* were prepared in the same fashion using the *Drosophila* genome DNA as a template for PCR amplification and primers provided in the Supplementary Table 1. Treatment of cells with dsRNA was performed over four days using a previously described protocol[48].

**Western-blot analysis.** Proteins were extracted with 8 M urea, 0.1 M Tris-HCl, pH 7.0, 1% SDS, fractionated by SDS-PAGE (12% acrylamide gel) and transferred to a PVDF membrane (Immobilon-P, Millipore). Blots were developed using alkaline phosphatase-conjugated secondary antibodies (Sigma) and the Immun-Star AP detection system (Bio-Rad). The following antibodies were used for detection: murine monoclonal anti-lamin Dm0 (1:2000; ADL67[49]), rabbit polyclonal anti-lamin C[25] (1:10000), murine monoclonal anti-beta Actin (1:3000; ab8224, Abcam).

**Chromatin visualisation by histone H4 or DAPI staining.** Lam-KD, LBR-KD or control S2 cells were seeded on coverslips for 30 min. After rinsing with PBS, cells were fixed in 100% methanol for 5 min at room temperature (for further examination of chromatin distribution based on the immunostaining of histone H4) or in 4% formaldehyde in PBS for 25 min at room temperature (for further estimation of chromatin volume based on DAPI staining), rinsed with PBS three times, blocked with PBTX (PBS with 0.1% Tween-20 and 0.3% Triton X-100) containing 3% normal goat serum (Invitrogen) for 1 h at room temperature. The remaining immunostaining procedure was performed as previously described[50]. As primary antibodies we used murine monoclonal anti-histone H4 (1:200; ab31830, Abcam), guinea pig polyclonal anti-LBR[26] (1:1000), rabbit polyclonal anti-lamin Dm0[51] (1:500). As the secondary antibodies we used Alexa Fluor 546-conjugated goat anti-rabbit IgG (Invitrogen) or Alexa Fluor 488-conjugated goat anti-mouse IgG (Invitrogen), or Alexa Fluor 633-conjugated goat anti-guinea pig IgG (Invitrogen).

**ImageJ quantitation of chromatin distribution.** Using ImageJ, we measured histone H4, LBR and lamin Dm0 profiles across the nucleus diameter of the equatorial focal plane of nuclei of Lam-KD, LBR-KD or control S2 cells. Fluorescent intensities were extracted, individual profiles were first normalised on the average intensity, then on the diameter of the nucleus (delimited by peaks of LBR fluorescence for Lam-KD, or by peaks of lamin Dm0 fluorescence for LBR-KD) and further aligned to determine the averaged profile. Nuclei from 2–3 independent experiments (60 nuclei per experiment) were analysed.

**Estimation of the volume of DAPI-stained chromatin.** Confocal images containing 20–30 DAPI-stained formaldehyde-fixed Lam-KD or control S2 cells were processed and analysed with the same parameters using IMARIS 7.4.2 software (Bitplane AG). Only nuclei with the lowest residual lamin Dm0 staining were used for analysis in Lam-KD cells, whereas in control cells, conversely, the nuclei with poor lamin Dm0 staining were not taken for analysis. For background subtraction, images were thresholded to ~15% of the maximal intensity of the channel so that the generated nuclear surfaces would not expand beyond the peak of LBR fluorescence intensity. With these parameters, the surfaces of nuclei, appropriate for analysis, were automatically reconstructed. Finally, the volumes of ~100 reconstructed nuclei were retrieved from the Statistics tab for the analysis.

**Two-colour FISH.** ~20-kb FISH probes were generated using a long-range PCR kit (Encyclo Plus PCR (Evrogen)) by PCR-amplification of 4 tiling genome fragments covering either the region 2 L:16964000–16982000 or 2 L:17310000–17328000, with the use of primer pairs provided in the Supplementary Table 1. 1 μg of template DNA for hybridization was labelled by random primed synthesis with the DIG DNA labelling kit (Roche) or by ChromaTide Alexa Fluor 546–14-dUTP (Life Technologies). Probes were further combined and hybridized with S2 cells as described previously[23]. For NL or FISH probe detection, as the primary antibodies we used guinea pig polyclonal anti-LBR[26] (1:1000), or rabbit polyclonal anti-lamin Dm0[51] (1:500) and sheep polyclonal anti-DIG-FITC (1:500, Roche). As the secondary antibodies we used Alexa Fluor 633-conjugated goat anti-guinea pig IgG (Invitrogen), or Alexa Fluor 546-conjugated goat anti-rabbit IgG (Invitrogen) and Alexa Fluor 488-conjugated goat anti-FITC IgG (Invitrogen).

**Measurement of distances between FISH probes and the NE.** Three-dimensional image stacks were recorded with a confocal LSM 510 Meta laser scanning microscope (Zeiss). Optical sections with 0.4-μm intervals along the Z-axis were captured. Images were processed and analysed by using IMARIS 7.4.2 software (Bitplane AG) with the blind experimental setup. Distances between both probes or between the probes and the NE were counted as previously described[23]. Briefly, we were unable to fully reconstruct the surfaces of nuclei automatically based on their LBR or lamin Dm0 immunostaining. Therefore, the nuclear rim of a particular nucleus was manually outlined in all optical sections of the stack by the middle of its LBR or lamin Dm0 staining to further reconstruct the surface of this nucleus automatically. To determine the distance between FISH signals and the NE, the instrument "measurement point" was positioned on the brightest voxel of the FISH probe and another "measurement point" was positioned on the reconstructed nuclear surface at the point of its earliest intersection with a progressively growing sphere from the first "measurement point". The distance between two "measurement points" (i.e. the shortest distance between the centre of the FISH probe and the middle of the NE) was measured for each nucleus. Distances between two FISH probes were measured correspondingly. Data were obtained in two independent experiments for 75–100 nuclei per experiment. In parallel, volumes of nuclei were retrieved, and radii of nuclei were calculated considering nuclei to be spherical. Finally, distances were normalised to the nuclear radii.

**Analysis of gene expression.** Total RNA was isolated from Lam-KD or control S2 cells using Trizol reagent (Invitrogen), and contaminating DNA was removed by DNase I treatment. RNA quality was assessed using capillary electrophoresis with a Bioanalyzer 2100 (Agilent). Poly(A)$^+$ RNA was extracted from total RNA using oligo(dT) magnetic beads (Thermo Fisher Scientific). NEBNext Ultra II RNA library preparation kit (New England Biolabs) was used for preparation of libraries following manufacturer's instructions. Libraries from two biological replicates of Lam-KD or control S2 cells were quantified using a Qubit fluorometer and quantitative PCR, and sequenced on the Illumina NextSeq resulting in 8.4–9.4 × $10^6$ 80-nt single-end reads. Reads were mapped to the *D. melanogaster* reference genome (version dm3) using HISAT[52] v2.1.0 with option –max-intronlen 50,000. Reads with low mapping quality were removed using SAMtools[53] with option -q 30. We calculated $\log_2$ transcription levels in 20-kb genomic bins using BEDtools[54] v2.16.2 with option -split, and then applied the hclust function in R to cluster the replicates using 1-Spearman's correlation coefficient as a distance metric. Gene expression was quantified with StringTie[52] for the reference annotation version r5.12. We filtered out genes with zero expression in more than two replicates. Among the remaining 10,076 genes, the differentially expressed genes were defined using the edgeR[55] package with trimmed mean of M values (TMM) normalisation at FDR = 0.05 cutoff. Genes were assigned to the LADs if their TSSs were located within LADs, while genes were assigned to the inter-LADs if their TSSs were at least 1-kb distant from LADs. A pseudocount was added to all expression values to get rid of zeros. The pseudocount was calculated as the minimal value in the gene expression table after normalisation. Then, we averaged the replicates and calculated $\log_2$(FC) values between Lam-KD and control samples.

Real-time RT-qPCR assay for the randomly selected genes from different LADs was performed on cDNAs synthesised with oligo(dT) primers on the poly(A)$^+$ RNA isolated from 3 biological replicates of Lam-KD or control S2 cells, using EvaGreen chemistry (Jena Bioscience) and the CFX96 hardware (BioRad). Expression levels of genes were normalised on the *act5C* gene expression. For semi-quantitative RT-PCR, applied for the analysis of genes from the 60D LAD, the reverse transcription of RNA was performed using SuperScript II reverse transcriptase (Invitrogen) in the presence of hexamer random primers. PCR amplification of cDNAs was performed with the addition of $^{33}$P-dATP. Probes after PCR were separated in 5% PAAG, which was then fixed, dried and exposed to the storage phosphor screen (Amersham Biosciences). The signals were scanned with a Phosphorimager Storm-820 (Molecular Dynamics). For each primer pair the number of PCR cycles were optimised to fit the exponential phase of amplification which was controlled by two-fold cDNA dilution. The expression levels of genes were normalised to the ubiquitous *CG4589* gene expression. Sequences of gene-specific primers are presented in the Supplementary Table 1.

**ChIP-seq procedure and data analysis.** ChIP-seq from two biological replicates of control and Lam-KD S2 cells with anti-H3-pan acetylated antibodies (Active Motif, #39139) was performed as previously described[56], with some modifications. After rinsing with PBS, ~2 × $10^7$ cells were fixed with 1.8% formaldehyde in PBS containing 0.5 mM DTT for 20 min at room temperature. Cross-linking was stopped by adding glycine to 225 mM for 5 min and washing in PBS containing 0.5 mM DTT three times for 5 min. Cells were washed once in the A2 buffer (140 mM NaCl, 15 mM HEPES pH 7.6, 1 mM EDTA, 0.5 mM EGTA, 1% Triton X-100, 0.1% sodium deoxycholate, 0.5 mM DTT, complete EDTA-free protease inhibitor cocktail (Roche)). Cells were lysed in the A2 buffer containing 1% SDS for 10 min at room temperature, after that the lysate was 20-fold diluted by the A2 buffer and incubated for 2 min at 4 °C. After sonication with VCX 400 Vibra-Cell Processor (Sonics; 30 pulses of 10 sec with 10-sec intervals at 15% max power) and 10-min high-speed centrifugation, the fragmented chromatin (with the average DNA fragment size ~0.5 kb) was recovered in the supernatant. For each immunoprecipitation, ~10 μg of chromatin (~700 μl) was pre-incubated in the presence of 100 μl of Protein A-Sepharose (PAS, 50% w/v, GE Healthcare) for 1 h at 4 °C. PAS was removed by centrifugation, 5% of chromatin was isolated as an "Input"

material, after that 2 μl anti-H3-pan acetylated antibodies (Active Motif, #39139) were added to the rest chromatin and samples were incubated overnight at 4 °C in a rotating wheel. Then, 100 μl of PAS was added and incubation was continued for 4 h at 4 °C. Samples were centrifuged at maximum speed for 1 min and the supernatant was discarded. Samples were washed four times in the A2 buffer containing 0.05% SDS and twice in 1 mM EDTA, 10 mM Tris (pH 8), 0.5 mM DTT buffer (each wash for 5 min at 4 °C). Chromatin was eluted from PAS in 100 μl of 10 mM EDTA, 1% SDS, 50 mM Tris (pH 8) at 65 °C for 10 min, followed by centrifugation and recovery of the supernatant. PAS material was re-extracted in 150 μl of TE, 0.67% SDS. To reverse cross-links, the combined eluate (250 μl) was incubated 6 h at 65 °C and treated by Proteinase K for 3 h at 50 °C. Samples were phenol-chloroform extracted and isopropanol precipitated in the presence of 20 μg glycogen. DNA was dissolved in 100 μl of water. ChIP samples containing ~25 ng of precipitated DNA, as well as "Input" samples were prepared for next-generation sequencing using a NEBNext Ultra II DNA library prep kit for Illumina (New England Biolabs). Libraries were sequenced on the Illumina HiSeq 2000 resulting in $3.1–3.4 \times 10^6$ 75-bp single-end reads. Reads were mapped to the *D. melanogaster* reference genome (version dm3) using Bowtie 2 v2.2.1 (with the –very-sensitive option)[57]. Reads with low mapping quality were removed using SAMtools[53] with option -q 30. Duplicate reads were removed using SAMtools rmdup. We calculated $\log_2$ ChIP and input signals in 1-kb genomic bins using BEDtools[54] v2.16.2, and then applied hclust function in R to cluster the replicates using 1-Spearman's correlation coefficient as a distance metric. Reads were assigned to LADs if they overlapped LADs, while reads were assigned to inter-LADs if they were at least 1-kb distant from LADs. We calculated read numbers within each LAD and inter-LAD, normalised the values for the sum of read coverage per replicate, excluded zero-covered LADs and inter-LADs from further analysis, averaged the replicates, and calculated $\log_2$(FC) values between Lam-KD and control ChIP samples.

**Hi-C procedure and data analysis.** Hi-C libraries from two independent biological replicates of control and Lam-KD S2 cells were prepared essentially as described previously[36] using the *HindIII-HF* restriction enzyme (NEB). Libraries were sequenced on the Illumina HiSeq 2000 platform resulting in $3–4 \times 10^7$ paired-end reads. Reads were mapped to the *D. melanogaster* reference genome (version dm3) using Bowtie 2 v2.2.1 (with the –very-sensitive option)[57]. The Hi-C data were processed using the ICE pipeline v0.9 (20 iterations of iterative correction) as described[58]. Hi-C interaction maps with 20-kb resolution were obtained. TADs were predicted using the Armatus software[59] v1.0, in which the average size and the number of TADs is determined by the scaling parameter γ. TAD annotation was performed in two steps as described[36]. First, we manually selected parameter γ to achieve good partitioning of TADs (γ = 1.20 for Lam-KD cells and γ = 1.12 for control cells). Then, TADs larger than 600 kb were split into smaller TADs with the scaling parameter γ multiplied by 2. After that, the smallest TADs (equal or less than 60 kb) were annotated as inter-TADs due to their poorly resolved internal structure. As a result, 576 (in control) and 588 (in Lam-KD) TADs were revealed. To examine whether TAD positions are altered upon Lam-KD, we analysed the degree of overlap of each TAD in the merged replicates of control and Lam-KD cells with that in the control replicates or in the Lam-KD replicates and did not find statistically significant difference ($P > 0.05$ in a two-sided Wilcoxon test). ACF within each TAD was calculated as an average value of iteratively corrected read numbers between all genomic bins belonging to the TAD, excluding boundary bins from both TAD sides. ACF within each inter-TAD was calculated as an average value of iteratively corrected read numbers between all genomic bins belonging to the inter-TAD and the boundary bins of adjacent TADs. For each TAD, we calculated the ratio between ACF value in each Lam-KD replicate and ACF value in each control replicate (four ratios in total). TADs with at least three ratios of the same sign were used for the downstream analysis. We note that when we selected TADs according to more strict criterion (i.e. all four ratios were changed in the same direction), it did not affect the results of analysis (Supplementary Fig. 6). Chromatin compartments were annotated using the principal component analysis as described[17]. Saddle plots were generated as described[58]. Briefly, we used the observed/expected Hi-C maps, which we calculated from 20-kb iteratively corrected interaction maps of *cis*-interactions by dividing each diagonal of a matrix by its chromosome-wide average value. In each observed/expected map, we rearranged the rows and the columns in the order of increasing PC1 values (which we calculated for the control matrices). Finally, we aggregated the rows and the columns of the resulting matrix into 20 equally sized aggregated bins, thus obtaining a saddle plot of compartmentalization.

**Analysis of published data.** We employed chromatin type annotation for S2 cells[40]. Proportions of chromatin types in 20-kb bins were calculated. Annotation of LADs was obtained from ref. [28]. We calculated the proportion of LAD length in each 20-kb TAD bin.

**Polymer modelling.** We used Dissipative Particle Dynamics (DPD) to perform computer simulations, as previously described[36] with some modifications. Briefly, macromolecules are represented in terms of the bead-and-spring model, with the particles interacting by a conservative force (repulsion), a dissipative force

(friction), and a random force (heat generator). A detailed description of the implementation of this technique was provided earlier[60]. The simulated cell volume was $50 \times 50 \times 50$ DPD units, density equals 3, so the total number of particles in the system is 375,000. We assume that a particle corresponds to a nucleosome. In addition, we introduce special boundary conditions, which are periodic for the solvent and impermeable for other particles. The surface consists of immobile, hexagonally positioned particles. In our simulations, particles mimic either "active" or "inactive" nucleosome types, while the surface mimics the NL. "Inactive" particles may create reversible "saturating" bonds[61,62] with each other as well as with the particles of a surface. Each "inactive" particle may have only one additional bond per moment, which simulates an interaction of a positively charged histone tail of non-acetylated nucleosome with the "acidic patch" of another nucleosome[42,43,63]. Our copolymer chain is represented by 64 blocks each consisting of 500 "inactive" and 50 "active" particles. The probability of creating an association between two "inactive" particles was set to 0.001, between the "inactive" particle and the surface – 0.007, while the probability to break such association was set to 0.01. During simulations, all particles were checked every 200 DPD steps, when the local equilibration was obtained. We performed 10 independent runs on the MSU supercomputer "Lomonosov-2" using our own implementation for the domain decomposition parallelised DPD code which is available at GitHub [https://github.com/KPavelI/dpd].

**Statistical analysis.** We applied the Wilcoxon test to check whether the distribution of $\log_2$(FC) values was symmetric around zero, as well as to test whether two distributions of $\log_2$(FC) values differed by a location shift of zero.

**Reporting Summary.** Further information on experimental design is available in the Nature Research Reporting Summary linked to this article.

## Data availability

Raw and processed Hi-C, RNA-seq and ChIP-seq data were deposited in the GEO NCBI under the accession number GSE110082. DPD code is available at GitHub [https://github.com/KPavelI/dpd]. The source data underlying Fig. 1a, b, d–f, h, i, 2d, 3, 5b, c and Supplementary Fig. 1c are provided as a Source Data file. All other data supporting the findings of this study are available from the corresponding authors upon request. A reporting summary for this Article is available as a Supplementary Information file.

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

## Acknowledgements

We thank Paul Fisher for anti-lamin C and anti-lamin Dm0 antibodies, Georg Krohne for anti-LBR antibody, Alexey A. Gavrilov (Faculty of Physics, M.V. Lomonosov Moscow State University) for DPD computer code. The research was carried out using the equipment of the shared research facilities of HPC computing resources at M.V. Lomonosov Moscow State University. This work was supported by the Russian Science Foundation (grant number 16–14–10081), by the Russian Foundation for Basic Research (grant number 17–00–00183) and by Foundation for the Advancement of Theoretical Physics "BASIS" (grant number 17–21–2101–1 to P.I.K.).

## Author contributions

S.V.U., S.A.D., E.E.K. and P.I.K contributed equally. Y.Y.S. and S.V.R. conceived the project. S.V.U. performed Hi-C and gene expression analysis. S.A.D. carried out lamin Dm0 and LBR depletions, western-blot analysis, ChIP, biological material preparation for RNA-seq, immunostaining and FISH experiments. E.A.M. maintained cell cultures. E.E.K. processed Hi-C data. E.E.K., S.V.U., A.A.I. and S.S.S. analysed various data sets (RNA-seq, ChIP-seq, Lamin DamID, chromatin type annotations). E.E.K., S.V.U., S.S.S. (supervised by E.E.K.), A.A.Galilsyna (supervised by M.S.G.) and I.M.F. performed Hi-C data analysis. V.V.N. performed FISH data analysis. P.I.K. and A.V.C.

performed polymer simulations. A.V.L. (supervised by A.A.Gavrilov) and P.I.K. performed analysis of polymer simulation data. A.V.L. (supervised by S.V.U.) performed PCA analysis. M.D.L. prepared libraries for ChIP-seq and RNA-seq, and carried out NGS of Hi-C, RNA-seq and ChIP-seq libraries. Y.Y.S. and S.V.U. wrote the manuscript with the input from all authors.

## Additional information

**Competing interests:** The authors declare no competing interests.

