## [Peer Review File · Nature Communications]

Reviewers' comments:

Reviewer #1 (Remarks to the Author):

In this manuscript, Ulianov and colleagues address the consequences of nuclear lamina disruption on chromatin conformation in *Drosophila* S2 cells. They show that the Lamin Dm0 knock-down results in a shift of the chromatin away from the nuclear periphery as well as a general compaction of the chromatin upon NL disruption. By performing HiC in control and Lam-KD cells, the authors make the surprising find that active chromatin regions become more compact while TADs that contain LADs show the opposite trend. The authors also show an increase in inter-TAD contacts between distinct chromatin states (active and silent) and between active TADs in Lam-KD cells. These results are further validated by the use of FISH and RT-qPCR. Finally, polymer modelling showed that chromatin attachment to the NL is responsible for the compaction of LADs. The authors conclude that the NL has an essential role in chromatin architecture by keeping LAD-containing TADs compact while allowing for active TADs to have a more stretched configuration.

The paper is in general well written, the results are novel and of interest to the field.

General points:

- A lot of the results and conclusions are based on the HiC data. It would be essential to benchmark their HiC data to previously released HiC data on S2 cells or another *Drosophila* cell line to validate the HiC results (eg Ramirez et al., 2016). The HiC data presented in Figure 2a does not look too convincing.
- The authors analyse data on LADs regardless of whether they are located in a TAD that has changed compaction level or not. It would be informative to separate these two groups of LADs (affected and unaffected) in a number of analyses. More specifically, this distinction could be used in finding specific chromatin states or gene expression that characterize only the affected LADs as well as to identify differential inter and intra-TAD contacts between the two classes of LADs.
- The S2 cell line is a male cell line and therefore contains an X-chromosome with a very particular chromatin state (namely H4K16ac) that could bias the analysis. The authors should consider either keeping this chromosome out of the analysis or process it separately from the remaining chromosomes.

Figure 1

Figure 1e – The FISH experiment provides a nice validation of the shift of LAD loci away from the NE but to have only one example is not sufficient. The use of more DNA-FISH probes would be important to show that this is a general effect caused by the Lam-KD.

Figure 1g - the authors conclude that depletion of LBR does not affect chromatin positioning. Although the entire chromatin doesn't seem to shift its location, it is possible that some loci do stray from the nuclear envelope. It would be important to show DNA-FISH with the same probe(s) as in figure 1e to further illustrate this point.

Figure 2

In order to fully characterize the different classes of TADs it would be essential to perform RNA-seq in lam-KD. This experiment will clarify how the changes in compaction differentially affect gene expression. Related to the above, it would also be interesting to perform Chip-seq of the relevant histone marks after NE disruption to investigate specific changes associated to DTADs and ITADs. The results shown in supplementary figure 2 are complementary to figure 2 but, for the same type of analysis, violin plots are shown in one figure and box plots in another. The type of plots should be consistent between the two figures.

Figure 2b - For the pie chart in this figure, the average contact frequency within the TAD is calculated. However, in some TADs there is a very clear mix of increased and decreased interactions. Such TADs can be classified as unchanged although significant changes occur within the TAD. These TADs, identified for example by taking into account the variability in contact change within one TAD, should be analysed in a separate class.

Figure 2f – Here the authors display the proportion of housekeeping genes in the different TAD types and find that there is a higher presence of such genes in ITADs. Housekeeping genes are used as an example of genes that expressed in these cells. However, it would be more informative to use an RNA-seq dataset instead, thereby including not just the housekeeping genes but also other expressed genes in S2 cells. The use of such a dataset would allow to analyse, not only the proportion of active genes within different classes of TADs, but also to calculate the proportion of low, medium and high expressed genes.

Figure 2g – in the y axis of the graph it should read: “LAD coverage within TAD”

Figure 3 – The authors look at interTAD contact changes and find increase in such contacts between active/silent and active/active TADs based on histone mark information. It could be more informative to use the classification of A and B compartments and analyse inter-TAD contacts between TADs from the same compartment or from different compartments.

Figure 4 – b) and c) are switched in the text

A FISH is performed with two probes at the edges of a LAD that gets shifted away from the NL. The authors should perform the same experiment with probes on a LAD that is unchanged in Lam-KD cells compared to the control.

Figure 4d – This panel displays gene-expression analysis in response to lam-KD by quantitative PCR. It is unclear whether the oligo-pairs for the genes in different LADs (top) or within the same LAD (bottom) are all regions within D-TADs or also contain regions in U-TADs. If these are only D-TAD genes –which I assume they are- it would be important to also select a panel of U-TAD genes to serve as a control panel. This analysis can also be performed with the RNA-seq proposed above.

Reviewer #2 (Remarks to the Author):

In this work, the authors studied the effects of the loss of nuclear lamina (NL) contacts on chromatin structure in *Drosophila* S2 cells. They use Hi-C, fluorescence microscopy, and FISH to characterize the consequences of knocking down (KD) the only lamin predominantly expressed in these cells. A statistically significant decrease in chromatin volume was observed, as well as changes in the compaction within about half of the TADs. Interestingly, some TADs were found to de-condense following this NL disruption, despite the overall genome-wide increased compaction of the chromatin. The authors then provide polymer simulations to show how such de-condensation could occur. Further, thinking that this de-condensation may lead to an elevated expression of genes within these TADs, the authors then provide data showing that the expression of some genes located within LADs was increased following NL disruption, although it was not clear if these genes are actually located within the TADs that de-condensed following the lamin KD.

Overall, this is a reasonably well written and thoroughly analyzed piece of work. However, while the statistical significance of the differences between control and lamin KD was demonstrated, the differences nonetheless frequently seem small. Overall, I think that the authors should provide some physically plausible justification for the potential functional relevance for global changes that are apparently so small.

My major concerns are:

1. The effect of the lamin KD on chromatin volume appears to be very small, which was surprising based on the title and abstract. From the shift in the histone H4 signal in Fig 1d, there is about a 4% decrease in radius. From the DAPI measurements in Fig 1f, there is a decrease of 3.6% in the median volume, corresponding to a 1.2% drop in radius. The latter corresponds to a 45 nm reduction in a nucleus whose median radius is 3750 nm and whose cell-to-cell, first-to-third quartile variation is 3520 nm to 4230 nm (Fig 1f). This 45 nm is roughly 1/15 of the thickness that LADs were found to occupy in the nuclear periphery in human cells (Cell 153, 178 (2013)). The authors' own threshold for determining whether or not a locus was attached to the NL is 15% of the nuclear radius (Fig 4b) or about 560 nm. And this small change in the radius of the chromatin volume is proposed to significantly change the compaction of ~50% of the TADs, including those that are not in direct contact with the NL (Fig 5e). How?

2. What is the justification for the 4% difference in fold-change (FC) in the average contact frequencies (ACFs) used for the determination of which TADs have significantly changed their compaction (ie, 0.96 and 1.04 in Fig 2b)? At first brush, I think this cutoff should depend on the FC variability between the two replicates themselves. Then, for example, the TADs categorized based on the differences between the replicate averages relative to this variability.

3. Would a 4% difference in ACF really be expected to be functionally relevant? With the normal cell population, cells at the first quartile have a 17% smaller volume and those at the third quartile have a 44% larger volume than the median volume. Perhaps the FC cutoff should be related to these differences. Or, a recent study (Nat Comm 8, 1753 (2017)) provided a quantitative relationship between Hi-C counts and the measured contact probability in *Drosophila*, and, except for those regions whose Hi-C counts are very low (and thus infrequently contact), a 4% change in ACF would be associated with, at most, a 5% change in contact probability. How could such a small global change to a TAD structure be significant functionally? Do the polymer simulations provide some insight?

4. Was the TAD studied with the two-color FISH in Fig 4a a TAD that de-condensed upon lamin KD (ie, a D-TAD)? Similarly, are all of the genes whose expression is higher after lamin KD in Fig 4c located within D-TADs? If not, together with the overall small changes in global structure, perhaps these changes in expression reflect only highly localized changes at these regions.

5. The authors argue that lamin KD results in a compaction of the genome rather than shrinkage of the nucleus as a whole. This is based on the observations of a change in volume from the DAPI measurements and no change in the LBR measurements (Fig 1f and Supplementary Fig 1d). However, the median volumes for the control and Lam-KD in the DAPI measurements are precisely the same as those in the LBR measurements. Likewise, they show precisely the same decrease. That is, using the median value as a characterization of the volume, the genome is compacted by the same amount as the nucleus shrinks. Also, in both the DAPI and the LBR measurements, there is a drop in the range of values observed in the control and the Lam-KD. I wonder if nuclear shrinkage contributes to the measured reduced chromatin volume. It is not clear why the range of values for the controls in the two cases is different. That is, given that there is apparently extensive contact of the chromatin with the NL in the control, shouldn't the volume measurement of the nucleus as a whole (by LBR) match precisely that of the chromatin volume (by DAPI)? Does DAPI have sufficient signal across the nucleus to reliably indicate chromatin volume? How about the volume from the Histone H4 images?

Minor points:

1. Supplementary Fig 2b shows that, for the D-TAD group, the intra-ACF of the control data set is statistically different from that of the Lam-KD data set. Similarly for the I-TADs, but not the U-TADs. However, the defining feature of each group is that the control is different from the Lam-KD

(for the D-TADs and I-TADs) or the same (for the U-TADs), and so shouldn't they necessarily be statistically different (or not) simply because of the way they were chosen into these groups? That is, say a similar selection process was performed just with the two control replicates – those TADs whose ACF is greater in control 1 than control 2 (group A), those with the same (group B), and those that are greater in control 2 than control 1 (group C). I suspect that there are many possible threshold values to make these groupings such that group A and group C would be statistically different but group B would not. But of course, the replicates are considered to be the same, with noise. So I think the comparison shown in Supplementary Fig 2b requires further explanation of its significance.

2. Why was the Kolmogorov-Smirnov (K-S) test used to evaluate the statistical significance of the differences in the FISH signals in Fig 1e, 4b and c? With all other comparisons between control and Lam-KD measurements, the Mann-Whitney U-test was employed.

3. Perhaps the authors could indicate in Fig 2a which TADs are D-TADs, I-TADs, and U-TADs. It appears that in the larger TADs, regions close to the diagonal are de-condensed (blue) whereas those further from the diagonal are condensed (orange). Since the U-TADs are in general larger than the other TADs, is this a general property of the U-TADs? Are the D-TADs and I-TADs in general more uniform in their de-condensation/condensation within the TAD? Perhaps size does contribute to their classification.

4. In the authors' earlier work (Genome Res 26, 70 (2016)), the authors noted that inter-TADs are enriched for active chromatin, similar to the I-TADs described here. Do the inter-TADs also become more condensed in the lamin KD?

5. On p. 4, the authors say that a certain % of the TADS increase/decrease their ACF "in both biological replicates". What does "in both biological replicates" mean here? Were all pair-wise sets compared?

6. A concern throughout this work: there seemed to be a difference between the sense given in the text of the differences between the control and Lam-KD measurements and the actual values associated with the differences. To be clear, the authors' description in the text accurately reflected the statistical differences observed. But, frequently, I thought that their description connoted something more absolute than what the data actually showed. For example, TADs "enriched" for active chromatin had a median of 10% active chromatin (Fig 2d) and are called "active TADs" (Fig 5e), while the TADs that were not enriched for active chromatin had a median of 5% active chromatin (Fig 2d) and are considered as inactive. Such a categorization with only a 5% difference does not appear to be functionally sound (particularly with the authors' own speculation of the structure of active TADs (Nucleus 7, 319 (2016))). Or on p.5, "both the D-TADs and the U-TADs strongly overlap with LADs which is not the case for I-TADs (Fig 2g)". The average proportion of LADs within TADs in the former pair is 42% and 49%, while that for the I-TAD is 28%. "Strongly overlap" is 42%, but 28% is essentially none (Fig 5e)?

7. The authors may wish to point out that the preservation of the TADs following lamin KD is consistent with the presence of equivalent TADs in polytene chromosomes (Cell 163, 934 (2015)).

8. Finally, a few typographic errors: in the text on p.6, Fig 4c was referred to as Fig 4b, and vice versa. Also, the label in Fig 2a indicates that the region is from Chr3R, while the legend says Chr3L.

Reviewer #3 (Remarks to the Author):

The report by Ulianov et al addresses the role of the nuclear periphery in organizing the 3D folding of the genome in a *Drosophila* cell line. This is an important topic as the nuclear periphery provides a sub-nuclear compartment that contributes to the separation of inactive from active chromatin domains. This process is believed to enable the emergence of robust phenotypes during development by permanently relocating genes active in immature cell states to the nuclear periphery while genes specifying the more mature state can be relocated from the periphery to undergo transcriptional activation in the nucleoplasm. The authors set out to examine these principles by exploring how the nuclear periphery might actually regulate 3D chromatin structures at a distance, ie in the interior of the nucleus. *Drosophila* cells used in this study, S2, lack Lamin C to facilitate the obliteration of the repressive nuclear peripheral compartment by knocking down expression of the remaining Lamin DmO. Unsurprisingly, this treatment led to a redistribution of the radial distribution of the genome. To gain further insight into the consequences of the loss of the chromatin-nuclear peripheral interaction, the authors used the Hi-C technique. This analysis revealed that the TADs distributed in the genome did not respond equally to prompt the identification of three TAD subtypes. The U-TADs, comprising the bulk of the TADs, showed no change, while D-TADs and I-TADs displayed more and less compacted chromatin, respectively, upon the knock down of Lamin DmO expression. Interestingly, the decompaction of LADs appeared to result in a modest although significant increase in transcriptional activity. Finally, the authors present computer modelling data suggesting that increased contacts with the nuclear lamina might be sufficient to compact LADs, to provide a rationale for the decompaction of TADs when these were released from the nuclear periphery.

I find this a timely and interesting study that provides experimental support for the perception that the nuclear periphery facilitates the compaction of chromatin. The experiments appear to have been expertedly implemented. The perhaps most interesting observation of this report is that the spatial separation of active and inactive domains was partially blurred in the absence of Lamin DmO. It is a pity therefore that the authors did not consider a ChIP-seq analysis to explore if this observation reflected changes in epigenomic profiles. The fact that the S2 cells transfected with dsRNA for Lamin DmO were maintained for four days would suggest considerable opportunities for replication-mediated propagation of changes in the epigenomic profiles.

Additional comments:

- I am negatively surprised that there is no mentioning of the degree of reproducibility of the replicas of the Hi-C data. This is key information for enabling an assessment of the quality of the data.
- There is no mentioning of the number of replicas for the western blot analysis of Fig. 1a. I also lack a statistical significance estimate of this data

Reviewer #1 (Remarks to the Author):

In this manuscript, Ulianov and colleagues address the consequences of nuclear lamina disruption on chromatin conformation in *Drosophila* S2 cells. They show that the Lamin Dm0 knock-down results in a shift of the chromatin away from the nuclear periphery as well as a general compaction of the chromatin upon NL disruption. By performing HiC in control and Lam-KD cells, the authors make the surprising find that active chromatin regions become more compact while TADs that contain LADs show the opposite trend. The authors also show an increase in inter-TAD contacts between distinct chromatin states (active and silent) and between active TADs in Lam-KD cells. These results are further validated by the use of FISH and RT-qPCR. Finally, polymer modelling showed that chromatin attachment to the NL is responsible for the compaction of LADs. The authors conclude that the NL has an essential role in chromatin architecture by keeping LAD-containing TADs compact while allowing for active TADs to have a more stretched configuration.

The paper is in general well written, the results are novel and of interest to the field.

General points:

Reviewer #1: A lot of the results and conclusions are based on the HiC data. It would be essential to benchmark their HiC data to previously released HiC data on S2 cells or another *Drosophila* cell line to validate the HiC results (eg Ramirez et al., 2016). The HiC data presented in Figure 2a does not look too convincing.

Our response: According to reviewer's suggestions we have compared our Hi-C results from S2 cells with those of Ramirez *et al.* (2015). The data sets appear to be highly correlated and this comparison (as well as correlation analysis of Hi-C replicates from our work) is included in the Supplementary Fig. 3a and 3b of the revised MS.

Reviewer #1: The authors analyse data on LADs regardless of whether they are located in a TAD that has changed compaction level or not. It would be informative to separate these two groups of LADs (affected and unaffected) in a number of analyses. More specifically, this distinction could be used in finding specific chromatin states or gene expression that characterize only the affected LADs as well as to identify differential inter and intra-TAD contacts between the two classes of LADs.

Our response: In the revised MS we have changed the principle of separation of TADs into groups. We divided TADs into four groups according to Jaccard coefficient calculated based on the proportion of LADs versus active chromatin in the TADs (Fig. 3d). For the polar TAD groups, we found opposed changes to their compaction upon Lam-KD. TADs with the high proportion of active chromatin and low LAD coverage increased their density, conversely TADs with a high LAD coverage and low proportion of active chromatin decreased their density upon Lam-KD (Fig. 3f). We think that TADs from intermediate groups B and C with unaltered or slightly changed density represent a mixture of active and inactive chromatin.

Reviewer #1: The S2 cell line is a male cell line and therefore contains an X-chromosome with a very particular chromatin state (namely H4K16ac) that could bias the analysis. The authors should consider either keeping this chromosome out of the analysis or process it separately from the remaining chromosomes.

Our response: We are grateful to the reviewer for this valuable comment. In the revised MS we analysed changes to ACF upon Lam-KD for the autosomes only, because we are not confident that LAD structure is the same on the female X chromosome in Kc167 cells and on the dosage-compensated male X chromosome in S2 cells. This explanation was added on p. 6 of the revised MS.

Reviewer #1: Figure 1e – The FISH experiment provides a nice validation of the shift of LAD loci away from the NE but to have only one example is not sufficient. The use of more DNA-FISH probes would be important to show that this is a general effect caused by the Lam-KD.

Our response: To provide more results with DNA-FISH probes, we reanalysed our previously published FISH data (Shevelyov et al. 2009) for two other loci (*22A* and *60D*). We now present the results from these probes as the radial-normalized distances (Fig. 1f). Of note, radial positions of these probes were shifted towards nuclear interior in response to Lam-KD. Moreover, total chromatin was also redistributed towards the nuclear interior (Fig. 1d). Collectively, these results strongly support our conclusion that total chromatin is shrunk upon NL disruption.

Reviewer #1: Figure 1g - the authors conclude that depletion of LBR does not affect chromatin positioning. Although the entire chromatin doesn't seem to shift its location, it is possible that some loci do stray from the nuclear envelope. It would be important to show DNA-FISH with the same probe(s) as in figure 1e to further illustrate this point.

Our response: We performed additional FISH experiments upon LBR-KD with the same probe as was used for Lam-KD and did not observe any notable changes in its position relative to the NE (Fig. 1i). This is in agreement with the absence of detectable shift in the position of total chromatin relative to the NE upon LBR-KD (Fig. 1h).

Reviewer #1: Figure 2. In order to fully characterize the different classes of TADs it would be essential to perform RNA-seq in lam-KD. This experiment will clarify how the changes in compaction differentially affect gene expression. Related to the above, it would also be interesting to perform Chip-seq of the relevant histone marks after NE disruption to investigate specific changes associated to DTADs and ITADs.

Our response: According to reviewer's suggestions, we performed RNA-seq, as well as ChIP-seq for the pan-acetylated histone H3 in control and Lam-KD S2 cells. We found that background transcription and H3 pan-acetylation are slightly increased in LADs, but are not notably altered in the inter-LADs upon NL disruption. Moreover, TADs carrying the largest proportion of LADs (group D according to our new classification) exhibit the highest, and TADs carrying the smallest proportion of LADs (group A) – the lowest increase of H3 pan-acetylation upon Lam-KD. Strikingly, transcription level increases in group D and decreases in group A upon Lam-KD. These results allow us to associate TADs which decrease their density upon Lam-KD with the high proportion of LADs inside them. These data were incorporated as Fig. 2a-c, and e (relative to LADs) and as Fig. 3f, g (relative to TADs) in the revised MS.

Reviewer #1: The results shown in supplementary figure 2 are complementary to figure 2 but, for the same type of analysis, violin plots are shown in one figure and box plots in another. The type of plots should be consistent between the two figures.

Our response: Violin plots were replaced with the box plots for uniformity.

Reviewer #1: Figure 2b - For the pie chart in this figure, the average contact frequency within the TAD is calculated. However, in some TADs there is a very clear mix of increased and decreased interactions. Such TADs can be classified as unchanged although significant changes occur within the TAD. These TADs, identified for example by taking into account the variability in contact change within one TAD, should be analysed in a separate class.

Our response: We calculated the dispersion of intra-TAD ACF changes for the four TAD groups and found that upon Lam-KD, TADs from groups A, B and C have higher variance of ACF changes than TADs from group D. These data support the idea that TADs from groups B and C represent a mixture of active and inactive chromatin which increases and decreases its density upon Lam-KD.

Reviewer #1: Figure 2f – Here the authors display the proportion of housekeeping genes in the different TAD types and find that there is a higher presence of such genes in ITADs. Housekeeping genes are used as an example of genes that expressed in these cells. However, it would be more informative to use an RNA-seq dataset instead, thereby including not just the housekeeping genes but also other expressed genes in S2 cells. The use of such a dataset would allow to analyse, not only the proportion of active genes within different classes of TADs, but also to calculate the proportion of low, medium and high expressed genes.

Our response: We removed the housekeeping genes from analysis, substituting them with the total transcription according to our RNA-seq data. The results of the analysis are presented on Fig. 3e of the revised MS.

Reviewer #1: Figure 2g – in the y axis of the graph it should read: “LAD coverage within TAD”

Our response: The X axis designation for Fig. 3b and Suppl. Fig. 3d, f is now the following: “LAD coverage” (due to space limitation the words “within TAD” were omitted).

Reviewer #1: Figure 3 – The authors look at interTAD contact changes and find increase in such contacts between active/silent and active/active TADs based on histone mark information. It could be more informative to use the classification of A and B compartments and analyse inter-TAD contacts between TADs from the same compartment or from different compartments.

Our response: We performed these analyses and incorporated the results as a separate text paragraph (p.7) and as the Fig. 4 of the revised MS.

Reviewer #1: Figure 4 – b) and c) are switched in the text

Our response: The content of b and c panels on Fig. 5 (previous Fig. 4) as well as the corresponding text in figure legend were interchanged.

Reviewer #1: A FISH is performed with two probes at the edges of a LAD that gets shifted away from the NL. The authors should perform the same experiment with probes on a LAD that is unchanged in Lam-KD cells compared to the control.

Our response: We now realize that upon Lam-KD TADs with an unaltered or slightly altered ACF (i.e. groups B and C) likely represent a mixture of active and inactive chromatin types (Fig. 3d and 3g). However, it is not easy to confirm this idea by FISH because the variance of intra-TAD ACF changes is rather high in these TADs and it will be necessary to capture weak ACF changes.

Reviewer #1: Figure 4d – This panel displays gene-expression analysis in response to lam-KD by quantitative PCR. It is unclear whether the oligo-pairs for the genes in different LADs (top) or within the same LAD (bottom) are all regions within D-TADs or also contain regions in U-TADs. If these are only D-TAD genes –which I assume they are- it would be important to also select a panel of U-TAD genes to serve as a control panel. This analysis can also be performed with the RNA-seq proposed above.

Our response: Upon Lam-KD, we revealed the derepression of transcription in the majority of LADs analysed (both by RT-qPCR and by RNA-seq), regardless of belonging of these LADs to the TAD of any particular group (Fig. 2b).

Reviewer #2 (Remarks to the Author):

In this work, the authors studied the effects of the loss of nuclear lamina (NL) contacts on chromatin structure in *Drosophila* S2 cells. They use Hi-C, fluorescence microscopy, and FISH to characterize the consequences of knocking down (KD) the only lamin predominantly expressed in these cells. A statistically significant decrease in chromatin volume was observed, as well as changes in the compaction within about half of the TADs. Interestingly, some TADs were found to de-condense following this NL disruption, despite the overall genome-wide increased compaction of the chromatin. The authors then provide polymer simulations to show how such de-condensation could occur. Further, thinking that this de-condensation may lead to an elevated expression of genes within these TADs, the authors then provide data showing that the expression of some genes located within LADs was increased following NL disruption, although it was not clear if these genes are actually located within the TADs that de-condensed following the lamin KD.

Overall, this is a reasonably well written and thoroughly analyzed piece of work. However, while the statistical significance of the differences between control and lamin KD was demonstrated, the differences nonetheless frequently seem small. Overall, I think that the authors should provide some physically plausible justification for the potential functional relevance for global changes that are apparently so small.

My major concerns are:

Reviewer #2: 1. The effect of the lamin KD on chromatin volume appears to be very small,

which was surprising based on the title and abstract. From the shift in the histone H4 signal in Fig 1d, there is about a 4% decrease in radius. From the DAPI measurements in Fig 1f, there is a decrease of 3.6% in the median volume, corresponding to a 1.2% drop in radius. The latter corresponds to a 45 nm reduction in a nucleus whose median radius is 3750 nm and whose cell-to-cell, first-to-third quartile variation is 3520 nm to 4230 nm (Fig 1f). This 45 nm is roughly 1/15 of the thickness that LADs were found to occupy in the nuclear periphery in human cells (Cell 153, 178 (2013)). The authors' own threshold for determining whether or not a locus was attached to the NL is 15% of the nuclear radius (Fig 4b) or about 560 nm. And this small change in the radius of the chromatin volume is proposed to significantly change the compaction of ~50% of the TADs, including those that are not in direct contact with the NL (Fig 5e). How?

Our response: We agree with the reviewer that effects of Lam-KD on chromatin density are not very strong, although they are statistically significant. Taking into account the reviewer's criticism we changed the title of the revised MS and made some changes in the text in order to reflect the fact that the influence of NL disruption on the 3D chromatin organization is not drastic.

Reviewer #2: 2. What is the justification for the 4% difference in fold-change (FC) in the average contact frequencies (ACFs) used for the determination of which TADs have significantly changed their compaction (ie, 0.96 and 1.04 in Fig 2b)? At first brush, I think this cutoff should depend on the FC variability between the two replicates themselves. Then, for example, the TADs categorized based on the differences between the replicate averages relative to this variability.

Our response: In the revised MS we rejected the cutoff for the fold change of increased or decreased contact frequencies, as it was introduced arbitrarily in the previous version of the MS. Now we analysed only changes in the intra-TADs ACF values that were the same in both Hi-C replicates. Also we have substantially changed the principle of separation of TADs into groups. We divided TADs into four groups according to the Jaccard coefficient calculated based on the LAD coverage versus active chromatin proportion in the TAD. For the polar TAD groups we found opposing changes in their compaction upon Lam-KD. TADs with the high proportion of active chromatin increase their density, whereas TADs with a high LAD coverage decrease their density upon Lam-KD. We think that TADs from the intermediate groups B and C, with unaltered or slightly changed density contain a mixture of active and inactive chromatin.

Reviewer #2: 3. Would a 4% difference in ACF really be expected to be functionally relevant? With the normal cell population, cells at the first quartile have a 17% smaller volume and those at the third quartile have a 44% larger volume than the median volume. Perhaps the FC cutoff should be related to these differences. Or, a recent study (Nat Comm 8, 1753 (2017)) provided a quantitative relationship between Hi-C counts and the measured contact probability in Drosophila, and, except for those regions whose Hi-C counts are very low (and thus infrequently contact), a 4% change in ACF would be associated with, at most, a 5% change in contact probability. How could such a small global change to a TAD structure be significant functionally? Do the polymer simulations provide some insight?

Our response: Changes in chromatin density upon Lam-KD appear to be functionally significant since our RNA-seq data demonstrate a total increase of the background

transcription in LADs as well as an altered expression of a number of genes located both in the LADs and in the nuclear interior.

Reviewer #2: 4. Was the TAD studied with the two-color FISH in Fig 4a a TAD that decondensed upon lamin KD (ie, a D-TAD)? Similarly, are all of the genes whose expression is higher after lamin KD in Fig 4c located within D-TADs? If not, together with the overall small changes in global structure, perhaps these changes in expression reflect only highly localized changes at these regions.

Our response: This is the TAD whose ACF is decreased upon Lam-KD (i.e. TAD from group D according to the new classification outlined in the response to the comment 2). Upon Lam-KD, we revealed the derepression of transcription in almost all the LADs analysed (both by RT-qPCR and by RNA-seq), regardless of which TAD group the LADs belonged to.

Reviewer #2: 5. The authors argue that lamin KD results in a compaction of the genome rather than shrinkage of the nucleus as a whole. This is based on the observations of a change in volume from the DAPI measurements and no change in the LBR measurements (Fig 1f and Supplementary Fig 1d). However, the median volumes for the control and Lam-KD in the DAPI measurements are precisely the same as those in the LBR measurements. Likewise, they show precisely the same decrease. That is, using the median value as a characterization of the volume, the genome is compacted by the same amount as the nucleus shrinks. Also, in both the DAPI and the LBR measurements, there is a drop in the range of values observed in the control and the Lam-KD. I wonder if nuclear shrinkage contributes to the measured reduced chromatin volume. It is not clear why the range of values for the controls in the two cases is different. That is, given that there is apparently extensive contact of the chromatin with the NL in the control, shouldn't the volume measurement of the nucleus as a whole (by LBR) match precisely that of the chromatin volume (by DAPI)? Does DAPI have sufficient signal across the nucleus to reliably indicate chromatin volume? How about the volume from the Histone H4 images?

Our response: Upon Lam-KD, volume of chromatin mass (by DAPI) is statistically significantly decreased, while the volume of nuclei (determined by LBR staining) is not significantly altered (although the median is slightly decreased). However, the main argument, supporting the idea that total chromatin is shrunk upon Lam-KD, and that this shrinkage is not mediated by the decrease of nucleus volume, is the redistribution of the radial position of chromatin towards the nuclear interior. This is supported by the radial positions of histone H4 (Fig. 1d) as well as by FISH probes (Fig. 1e, f). These data clearly indicate that a gap between the surface of chromatin and the NE becomes larger upon NL disruption.

DAPI stains the whole volume of chromatin as could be seen from Supplementary Fig. 1c. The results of chromatin volume measurements by H4 or DAPI staining are different because nuclei were fixed with methanol for H4 immunostaining, or with formaldehyde in case of DAPI and LBR measurements.

Minor points:

Reviewer #2: 1. Supplementary Fig 2b shows that, for the D-TAD group, the intra-ACF of the control data set is statistically different from that of the Lam-KD data set. Similarly for

the I-TADs, but not the U-TADs. However, the defining feature of each group is that the control is different from the Lam-KD (for the D-TADs and I-TADs) or the same (for the U-TADs), and so shouldn't they necessarily be statistically different (or not) simply because of the way they were chosen into these groups? That is, say a similar selection process was performed just with the two control replicates – those TADs whose ACF is greater in control 1 than control 2 (group A), those with the same (group B), and those that are greater in control 2 than control 1 (group C). I suspect that there are many possible threshold values to make these groupings such that group A and group C would be statistically different but group B would not. But of course, the replicates are considered to be the same, with noise. So I think the comparison shown in Supplementary Fig 2b requires further explanation of its significance.

Our response: In the revised MS we analysed only changes in the intra-TADs ACF values that were the same in both Hi-C replicates. We also changed the principle of separation of TADs into groups. We divided TADs into four groups according to the Jaccard coefficient calculated based on the proportion of LADs versus active chromatin in the TADs. For the polar TAD groups we found opposing changes in their compaction upon Lam-KD. TADs with the high proportion of active chromatin increase their density, whereas TADs with the high LAD coverage do oppositely decrease their density upon Lam-KD. We think that TADs from the intermediate groups B and C, with unaltered or slightly changed density contain a mixture of active and inactive chromatin.

Reviewer #2: 2. Why was the Kolmogorov-Smirnov (K-S) test used to evaluate the statistical significance of the differences in the FISH signals in Fig 1e, 4b and c? With all other comparisons between control and Lam-KD measurements, the Mann-Whitney U-test was employed.

Our response: All statistical analysis for the positions of FISH signals was redone by Kolmogorov-Smirnov test in the revised MS. We applied the KS test because it is sensitive to any differences in the two distributions, including differences in shape, spread or median, while Wilcoxon (or Mann-Whitney) test is sensitive to the median changes only.

Reviewer #2: 3. Perhaps the authors could indicate in Fig 2a which TADs are D-TADs, I-TADs, and U-TADs. It appears that in the larger TADs, regions close to the diagonal are de-condensed (blue) whereas those further from the diagonal are condensed (orange). Since the U-TADs are in general larger than the other TADs, is this a general property of the U-TADs? Are the D-TADs and I-TADs in general more uniform in their de-condensation/condensation within the TAD? Perhaps size does contribute to their classification.

Our response: We agree with the reviewer that changes in the ACF for a particular TAD may represent a mixture of different changes within it. To address this at the genome-wide scale, we calculated the variance of intra-TAD ACF changes for the TADs from four groups defined as described above (answer to the minor point 1). We found that upon Lam-KD, TADs from groups A, B and C have higher variance of ACF changes than TADs from group D. These data support the idea that TADs from groups B and C represent a mixture of active and inactive chromatin which increases and decreases its density upon Lam-KD.

Reviewer #2: 4. In the authors' earlier work (Genome Res 26, 70 (2016)), the authors noted

that inter-TADs are enriched for active chromatin, similar to the I-TADs described here. Do the inter-TADs also become more condensed in the lamin KD?

Our response: We performed the analysis of changes to ACF in the inter-TADs, recommended by reviewer, and found that they, indeed, become more condensed upon Lam-KD (Fig. 3h).

Reviewer #2: 5. On p. 4, the authors say that a certain % of the TADs increase/decrease their ACF “in both biological replicates”. What does “in both biological replicates” mean here? Were all pair-wise sets compared?

Our response: “In both biological replicates” means that at least 3 out of 4 pair-wise ratios of Lam-KD to control should be changed in the same direction (i.e. increased or decreased).

Reviewer #2: 6. A concern throughout this work: there seemed to be a difference between the sense given in the text of the differences between the control and Lam-KD measurements and the actual values associated with the differences. To be clear, the authors’ description in the text accurately reflected the statistical differences observed. But, frequently, I thought that their description connoted something more absolute than what the data actually showed. For example, TADs “enriched” for active chromatin had a median of 10% active chromatin (Fig 2d) and are called “active TADs” (Fig 5e), while the TADs that were not enriched for active chromatin had a median of 5% active chromatin (Fig 2d) and are considered as inactive. Such a categorization with only a 5% difference does not appear to be functionally sound (particularly with the authors’ own speculation of the structure of active TADs (Nucleus 7, 319 (2016))). Or on p.5, “both the D-TADs and the U-TADs strongly overlap with LADs which is not the case for I-TADs (Fig 2g)”. The average proportion of LADs within TADs in the former pair is 42% and 49%, while that for the I-TAD is 28%. “Strongly overlap” is 42%, but 28% is essentially none (Fig 5e)?

Our response: According to reviewer’s comments we made text corrections in order to describe the obtained results more accurately. In particular, we removed from the text such definitions as “active” or “inactive” TADs and employed the following characteristics: “TADs containing the highest proportion of active chromatin (group A)” or “TADs highly corresponding to LADs (group D)”

Reviewer #2: 7. The authors may wish to point out that the preservation of the TADs following lamin KD is consistent with the presence of equivalent TADs in polytene chromosomes (Cell 163, 934 (2015)).

Our response: We have cited this publication in the revised MS.

Reviewer #2: 8. Finally, a few typographic errors: in the text on p.6, Fig 4c was referred to as Fig 4b, and vice versa. Also, the label in Fig 2a indicates that the region is from Chr3R, while the legend says Chr3L.

Our response: These text errors were fixed.

Reviewer #3 (Remarks to the Author):

The report by Ulianov et al addresses the role of the nuclear periphery in organizing the 3D folding of the genome in a *Drosophila* cell line. This is an important topic as the nuclear periphery provides a sub-nuclear compartment that contributes to the separation of inactive from active chromatin domains. This process is believed to enable the emergence of robust phenotypes during development by permanently relocating genes active in immature cell states to the nuclear periphery while genes specifying the more mature state can be relocated from the periphery to undergo transcriptional activation in the nucleoplasm. The authors set out to examine these principles by exploring how the nuclear periphery might actually regulate 3D chromatin structures at a distance, ie in the interior of the nucleus. *Drosophila* cells used in this study, S2, lack Lamin C to facilitate the obliteration of the repressive nuclear peripheral compartment by knocking down expression of the remaining Lamin DmO. Unsurprisingly, this treatment led to a redistribution of the radial distribution of the genome. To gain further insight into the consequences of the loss of the chromatin-nuclear peripheral interaction, the authors used the Hi-C technique. This analysis revealed that the TADs distributed in the genome did not respond equally to prompt the identification of three TAD subtypes. The U-TADs, comprising the bulk of the TADs, showed no change, while D-TADs and I-TADs displayed more and less compacted chromatin, respectively, upon the knock down of Lamin DmO expression. Interestingly, the decompaction of LADs appeared to result in a modest although significant increase in transcriptional activity. Finally, the authors present computer modelling data suggesting that increased contacts with the nuclear lamina might be sufficient to compact LADs, to provide a rationale for the decompaction of TADs when these were released from the nuclear periphery.

Reviewer #3: I find this a timely and interesting study that provides experimental support for the perception that the nuclear periphery facilitates the compaction of chromatin. The experiments appear to have been expertedly implemented. The perhaps most interesting observation of this report is that the spatial separation of active and inactive domains was partially blurred in the absence of Lamin DmO. It is a pity therefore that the authors did not consider a ChIP-seq analysis to explore if this observation reflected changes in epigenomic profiles. The fact that the S2 cells transfected with dsRNA for Lamin DmO were maintained for four days would suggest considerable opportunities for replication-mediated propagation of changes in the epigenomic profiles.

Our response: According to reviewer's suggestions, we performed ChIP-seq for the pan-acetylated histone H3 in control and Lam-KD S2 cells. We found that H3 pan-acetylation was clearly increased in LADs, but was not notably altered in the inter-LADs upon NL disruption. We also found that upon Lam-KD, H3 pan-acetylation is mostly increased in the TADs from group D which appear to be decondensed upon Lam-KD. These data were incorporated as Fig. 2e and Fig. 3f in the revised MS.

Additional comments:

Reviewer #3: • I am negatively surprised that there is no mentioning of the degree of reproducibility of the replicas of the Hi-C data. This is key information for enabling an assessment of the quality of the data.

Our response: The Pearson's correlation coefficients between Hi-C replicates are now presented on the Supplementary Fig. 3b. The correlation between replicates is very high. Moreover, as a benchmark, we included the comparison of our Hi-C data on the S2 cells with that previously published (Supplementary Fig. 3a; Ramirez *et al.* 2015).

Reviewer #3: • There is no mentioning of the number of replicas for the western blot analysis of Fig. 1a. I also lack a statistical significance estimate of this data.

Our response: All experiments contained at least two replicates and the error bars were included in Fig. 1a of the revised MS.

Reviewers' comments:

Reviewer #1 (Remarks to the Author):

The manuscript by Ulianov., et al significantly increased in quality and gained appeal upon the incorporation of almost all the reviewers comments. The work is solid, their findings are novel and interesting to the field and the text appropriately reflects the outcomes. I recommend publication in Nature Communications.

Reviewer #2 (Remarks to the Author):

In this revised work, the authors have satisfactorily addressed my previous concerns. However, I have some new concerns with their newly included data and analysis.

1. The new method of determining whether or not a TAD is included in the subsequent analysis lacks statistical justification, as it is simply whether or not three of the four possible average contact frequency (ACF) ratios (Control 1/KD1, Control 1/KD2, Control 2/KD1, and Control 2/KD2) are all >1 or all <1 . But that, of course, does not mean it is "wrong". I tend to agree with the authors that the statistical significance of the properties (differences in intra-ACF, transcription, etc) of the sub-groups A and D of this chosen dataset do strongly suggest biologically meaningful differences upon KD. Perhaps the authors could apply a more stringent test – "all four of the ratios are >1 or <1 " or "three of the four are all >1.02 or all <1.02 " – and then determine whether this group shows the same trend as that obtained with their original criteria. This more strictly chosen group might be too small to enable the same statistical analyses as the original, but if the trend persists, this information could be included in the manuscript to provide further confidence in their analysis, even in the form as "data not shown".
2. The results shown in Figure 2a do not appear to be consistent with the designation of the black and red genes as the most "differentially expressed" (DE) in their data. There are clearly several "not differentially expressed" genes (blue and grey) that are at precisely the same (perpendicular) distance from the dotted line, and many at a greater distance, than the red and black genes.
3. There are things wrong with Supplementary Figure 1: the legend to (b) is precisely the same as (a) but the data are clearly different; the legend to (d) might describe the results shown in (b); and it is not clear if the data in (a) is supposed to show something different than (d).
4. On p. 18, the authors indicate that they applied the Wilcoxon test to determine whether the data was symmetric around zero. Perhaps they could include more details about this calculation. Did they generate the "negative" of their distribution and then examine the "positive" and "negative" distributions with the Wilcoxon?
5. A minor concern – while the greater variances of groups B and C than of group D shown in Fig 3g is consistent with the former consisting of a mixture of active and inactive chromatin types as the authors suggest, why is the variance of group A, which is apparently a more "pure" group than B or C, the greatest of the four?
6. Finally, the authors could also indicate the correlation between the biological replicates in the RNA-seq data as well as the total number of TADs in their Hi-C data (prior to applying the criteria described in comment #1). Also, is the P value shown in Fig 1d associated with just the H4 data (and not the LBR data)? If so, this should be indicated in the legend.

Reviewer #2 (Remarks to the Author):

In this revised work, the authors have satisfactorily addressed my previous concerns. However, I have some new concerns with their newly included data and analysis.

Our response: We thank the Reviewer for the thorough analysis of our work. We believe that now we have addressed all new comments of the Reviewer.

1. The new method of determining whether or not a TAD is included in the subsequent analysis lacks statistical justification, as it is simply whether or not three of the four possible average contact frequency (ACF) ratios (Control 1/KD1, Control 1/KD2, Control 2/KD1, and Control 2/KD2) are all >1 or all <1 . But that, of course, does not mean it is "wrong". I tend to agree with the authors that the statistical significance of the properties (differences in intra-ACF, transcription, etc) of the sub-groups A and D of this chosen dataset do strongly suggest biologically meaningful differences upon KD. Perhaps the authors could apply a more stringent test – "all four of the ratios are >1 or <1 " or "three of the four are all >1.02 or all <1.02 " – and then determine whether this group shows the same trend as that obtained with their original criteria. This more strictly chosen group might be too small to enable the same statistical analyses as the original, but if the trend persists, this information could be included in the manuscript to provide further confidence in their analysis, even in the form as "data not shown".

Our response: We performed the analysis suggested by the Reviewer (i.e. all four of the ratios are >1 or <1). The obtained results are very similar to the ones presented in the current version of the manuscript when three of the four combinations $>$ or <1 were considered. This information was incorporated in the revised manuscript as the Supplementary Figure 6 and as the sentence on p. 16: "We note that when we selected TADs according to more strict criterion (i.e. all four ratios were changed in the same direction), it did not affect the results of analysis (Supplementary Fig. 6)."

2. The results shown in Figure 2a do not appear to be consistent with the designation of the black and red genes as the most "differentially expressed" (DE) in their data. There are clearly several "not differentially expressed" genes (blue and grey) that are at precisely the same (perpendicular) distance from the dotted line, and many at a greater distance, than the red and black genes.

Our response: As stated in the Methods section on p. 14, we define differentially expressed (DE) genes "using the edgeR package with trimmed mean of M values (TMM) normalisation at FDR = 0.05 cut-off". Accordingly, the designation of DE genes (black and red) in Figure 2a is based on this definition. Genes are highlighted as DE only if they demonstrate FDR < 0.05 . As the Reviewer has pointed out correctly, there are several "not DE" genes (blue and grey) at the greater (perpendicular) distance from the dotted line than the DE genes. These particular "not DE" genes demonstrate FDR ≥ 0.05 due to the variations between replicates.

3. There are things wrong with Supplementary Figure 1: the legend to (b) is precisely the same as (a) but the data are clearly different; the legend to (d) might describe the results shown in (b); and it is not clear if the data in (a) is supposed to show something different than (d).

Our response: The legends to Supplementary Figures 1b and 1d were accidentally mixed up. Now it is corrected.

4. On p. 18, the authors indicate that they applied the Wilcoxon test to determine whether the data was symmetric around zero. Perhaps they could include more details about this calculation. Did they generate the "negative" of their distribution and then examine the "positive" and "negative" distributions with the Wilcoxon?

Our response: Our statement on p. 18 simply indicated that the Wilcoxon test was used, in a conventional way, for two null hypotheses: (1) the null hypothesis is that the distribution of the sample is symmetric around zero; (2) the null hypothesis is that the distributions of sample 1 and sample 2 differ by a location shift of zero and the alternative is that they differ by some other location shift. We have now added a clarification to the Methods section (at p. 18): "We applied the Wilcoxon test to check whether the distribution of $\log_2(\text{FC})$ values was symmetric around zero, as well as to

test whether two distributions of $\log_2(\text{FC})$ values differed by a location shift of zero.”

5. A minor concern – while the greater variances of groups B and C than of group D shown in Fig 3g is consistent with the former consisting of a mixture of active and inactive chromatin types as the authors suggest, why is the variance of group A, which is apparently a more “pure” group than B or C, the greatest of the four?

Our response: The increased variance of group A might be due to the presence of a notable fraction of LADs mixed with the largest fraction of active chromatin in group A TADs. Another possible interpretation is that active chromatin per se could increase variance in intra-TAD ACF due to the increased flexibility of the nucleosome fiber in the open chromatin regions. We have now changed the sentence at p. 7 to the following: “In support of this idea, the variance of ACF changes is the lowest within group D TADs (Fig. 3g) which strongly corresponds to LADs, when compared to other groups containing the mixture of active and inactive chromatin types (Fig. 3d)”

6. Finally, the authors could also indicate the correlation between the biological replicates in the RNA-seq data as well as the total number of TADs in their Hi-C data (prior to applying the criteria described in comment #1). Also, is the P value shown in Fig 1d associated with just the H4 data (and not the LBR data)? If so, this should be indicated in the legend.

Our response: According to the Reviewer’s suggestions, Spearman’s correlation coefficients between RNA-seq and ChIP-seq replicates were indicated on the Supplementary Figures 2a and 2d. Information about TAD numbers is now included at p. 16 of the revised manuscript. The indication that P-value is attributed to the H4 profiles is now incorporated in Figure 1d legend of the revised manuscript.

REVIEWERS' COMMENTS:

Reviewer #2 (Remarks to the Author):

The authors have satisfactorily addressed my concerns.

Reviewer #2 (Remarks to the Author):

The authors have satisfactorily addressed my concerns.

Our response: We thank the Reviewer for the thorough analysis of our work.